# Network state changes in sensory thalamus represent learned outcomes

Masashi Hasegawa [1,2,4], Ziyan Huang [1,4], Ricardo Paricio-Montesinos [1] & Jan Gründemann [1,2,3] ✉

Thalamic brain areas play an important role in adaptive behaviors. Nevertheless, the population dynamics of thalamic relays during learning across sensory modalities remain unknown. Using a cross-modal sensory reward-associative learning paradigm combined with deep brain two-photon calcium imaging of large populations of auditory thalamus (medial geniculate body, MGB) neurons in male mice, we identified that MGB neurons are biased towards reward predictors independent of modality. Additionally, functional classes of MGB neurons aligned with distinct task periods and behavioral outcomes, both dependent and independent of sensory modality. During non-sensory delay periods, MGB ensembles developed coherent neuronal representation as well as distinct co-activity network states reflecting predicted task outcome. These results demonstrate flexible cross-modal ensemble coding in auditory thalamus during adaptive learning and highlight its importance in brain-wide cross-modal computations during complex behavior.

Relaying sensory information from subcortical areas to sensory cortices is a fundamental function of thalamus[1,2], yet we are only starting to understand the computational role of individual sensory thalamic nuclei in flexible encoding during adaptive behaviors[3–7]. Auditory thalamus (medial geniculate body, MGB) flexibly encodes sensory information across associative learning[8–12], and populations of individual MGB neurons exhibit diverse single cell response adaptations to conditioned tone stimuli in a behavioral state-dependent manner during fear learning[5]. In addition to learning-related auditory response plasticity, MGB neurons process cross-modal sensory inputs in a complex manner[10,13]. For example, visual stimuli enhance MGB tone responses non-linearly[10], while tactile stimuli affect MGB tone responses bidirectionally[13], indicating that, similar to cortical and collicular brain areas, MGB processes cross-modal sensory inputs in addition to its auditory relay function[14–17]. These data suggest that MGB is an active computational unit which processes complex information across sensory modalities upon adaptive behaviors. Nevertheless, it remains unknown how large-scale neuronal dynamics in auditory thalamus represent sensory stimuli of different modalities that change their assigned value and expected outcome during flexible learning.

Here, we show that sensory thalamus dynamically encodes sensory as well as task-related information, adapting its neural dynamics and network state to varying task rules.

## Results

### Auditory thalamus neurons display various response patterns to cross-modal sensory stimuli in reward associative learning

We developed a cross-modal sensory Go/Nogo reversal learning task in mice to test for neuronal population dynamics during sensory learning and for cognitive flexibility upon changing reward contingencies. Mice ($N = 8$ animals) were trained to associate counterbalanced auditory (12 kHz tone) and visual (rightward drifting grating) stimuli as Go (reward predictor) or Nogo (not rewarded) cues (Fig. 1a–c). Once mice learned the task at expert level (*Initial learning*), the reward contingency was reversed and the previous Go cue became non-predictive of the reward, while the previous Nogo cue turned into a reward-predictor (*Reversal learning*). Animals learned the initial rule within ~10 days (Figs. 1d–f and Supplementary Fig. 1). Upon reversal of the reward contingencies, task performance dropped initially, yet mice learned the new stimulus-reward rule again with a similar learning rate ($N = 8$

[1]German Center for Neurodegenerative Diseases (DZNE), Neural Circuit Computations, Bonn, Germany. [2]Department of Biomedicine, University of Basel, Basel, Switzerland. [3]University of Bonn, Faculty of Medicine, Bonn, Germany. [4]These authors contributed equally: Masashi Hasegawa, Ziyan Huang. ✉e-mail: jan.grundemann@dzne.de

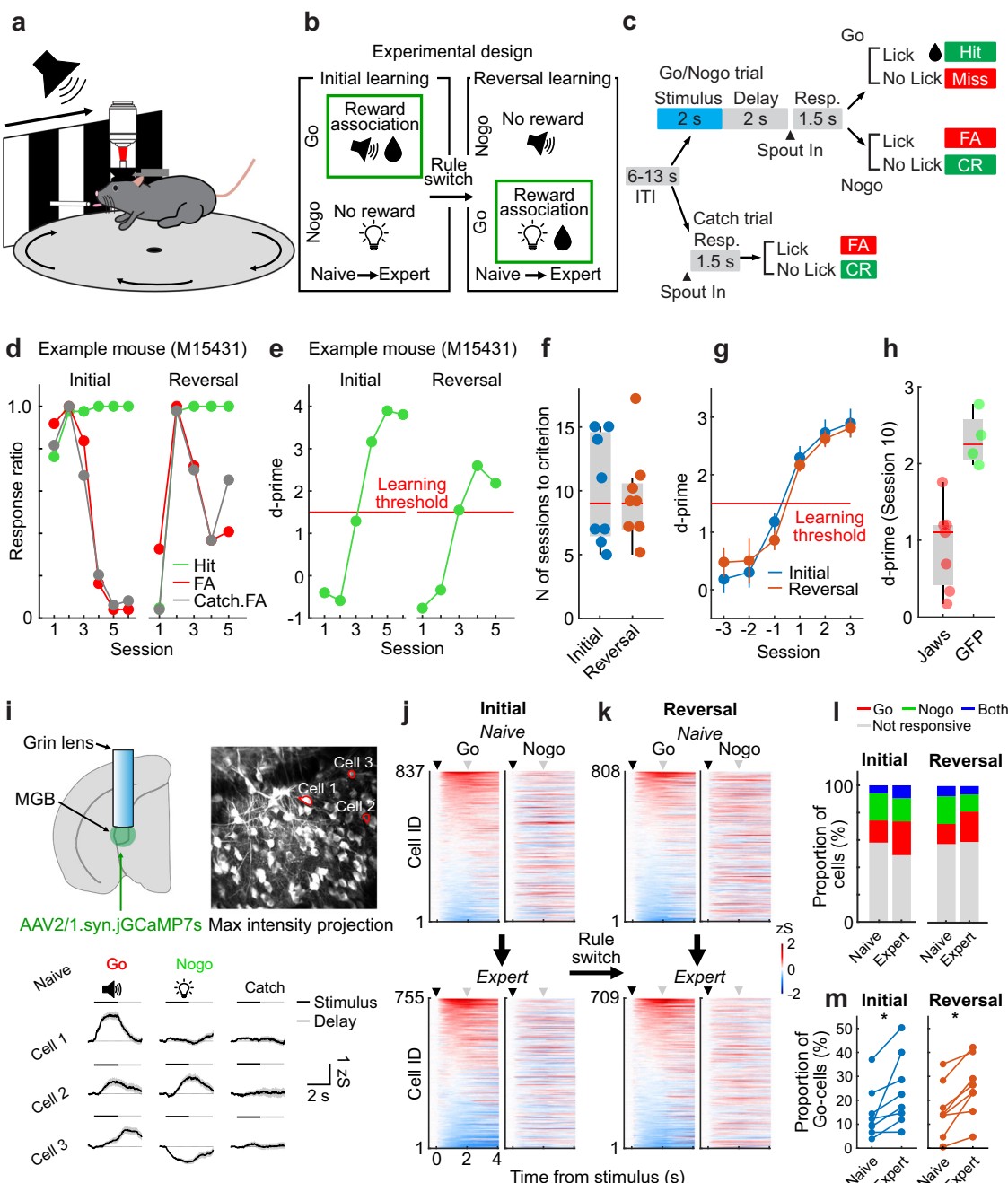

**Fig. 1 | Auditory thalamus neurons exhibit diverse responses and dynamic bias towards reward predictors upon cross-modal reversal learning. a** Behavioral setup schematic. **b** Design of the Go/Nogo reversal learning paradigm. **c** Trial structure, response types and their outcomes. Hit: correct lick and reward delivery. Miss: no lick to Go stimulus. False Alarm (FA): lick to Nogo stimulus. Correct Rejection (CR): no-lick to Nogo/Catch. Miss and FA are followed by timeouts. **d, e** Performance and d-prime transition of one example mouse. Threshold: d-prime > 1.5. d-prime is calculated from Hit and FA responses. **f** Number of sessions to learning criterion ($N = 8$ mice). Stimulus-reward association sequence (auditory→visual vs. visual→auditory) is counterbalanced across mice ($N = 4$ for each group). **g** d-prime transition around learning threshold are similar between initial and reversal learning (mean ± SEM, $N = 8$ mice). **h** d-prime values of Jaws ($N = 7$) and GFP ($N = 4$) groups on Day 10. **i** Lens implantation scheme and example two-photon max. intensity projection of MGB (enhanced contrast visualization). Bottom: Ca²⁺ traces from example neurons in distinct trials (mean ± SEM). **j, k** Single cell activity

in Go and Nogo trials during initial and reversal learning ($N = 8$ mice) sorted by the mean stimulus amplitude of Go trials. Cell IDs are matched across Go and Nogo trials. Black and gray triangles represent the stimulus and delay period onset. **l** Proportion of stimulus-responsive cells from the data shown in (**j** and **k**). The transition of the proportion of stimulus-responsive cells from naive to expert phases in the initial learning (left) and reversal learning (right). The proportions of the stimulus-responsive cells were altered from Naive to Expert phases in both initial and reversal learning (both $p < 0.0001$, Chi-square, two-sided). **m** Proportion of Go-responsive cells in naive and expert phases in initial and reversal learning ($N = 8$ mice). The proportion of Go-responsive cells increased from naive to expert (both $p = 0.0156$, sign-rank test, two-sided). Boxplots in Fig. 1f, h show median, lower and upper quartiles (box edges), maximum and minimum values without outliers. Values beyond the 1.5*interquartile range from the lower or upper quartiles are considered outliers.

mice, Figs. 1d–g and Supplementary Fig. 1). Optogenetic inhibition of MGB activity perturbed the task learning (Fig. 1h and Supplementary Fig. 2). These data indicate that mice flexibly associate sensory stimuli with reward outcome across initial and reversal learning and MGB is required for acquisition of the task.

To track the activity patterns of MGB neurons during learning, we performed longitudinally in vivo two-photon calcium imaging of large populations of individual MGB neurons through a gradient refractive index (GRIN) lens across all stages of the learning paradigm (Fig. 1i, Supplementary Movie 1). At first, we investigated how MGB neurons responded to cross-modal sensory stimuli and how their responses were altered across reward associative learning by analyzing all recorded neurons, including longitudinally tracked cells and non-tracked cells. During naive and expert phases of initial and reversal learning, MGB neurons exhibited a large variety of distinct response patterns to the auditory tone as well as the visual stimulus and combinations thereof (Fig. 1i–k, Supplementary Fig. 3), indicating that subsets of MGB neurons are inherently responsive to cross-modal sensory stimuli. Nevertheless, the learning-induced reward association of the Go stimulus during initial or reversal learning altered the proportion of stimulus-responsive MGB neurons towards the Go stimulus regardless of the sensory modality (Figs. 1l, m, Supplementary Fig. 3). These results demonstrate that auditory thalamus processes cross-modal sensory information during discriminatory reward learning and that MGB responsiveness is dynamically biased towards reward predictors independent of stimulus modality.

### Functional neuronal subgroups predict task-outcome in MGB
Next, we separated MGB neurons into functional subgroups by using a k-means-based cluster analysis approach using longitudinally tracked cells (Supplementary Fig. 4a). MGB neurons exhibited distinct stable, learning-enhanced as well as learning-inhibited responses to the reward-predicting Go stimuli (Fig. 2a, b). These learning-related functional clusters in MGB emerged regardless of the modality of the Go stimulus (Fig. 2a, b, right). In addition to stimulus-driven responses, subsets of MGB neurons developed ramping activity during the non-sensory, pre-reward delay period of Go trials (Ramp-up or Ramp-down clusters), during which the animal has to retain the stimulus type (Go vs. Nogo) and prepare for the action. This ramping activity was specific to the Go stimulus, given that only a small proportion of neurons exhibited ramping activity during Nogo delay periods (Supplementary Fig. 4b-f). Furthermore, ramping activity was only observed in Hit trials, not FA trials, indicating that this activity is correlated with task outcome (Supplementary Fig. 5).

We then compared the activity patterns of the same neurons across expert states in initial and reversal learning when the animals are successfully performing the task and found task-specific MGB neurons that are active during distinct trial epochs irrespective of the sensory modality (Fig. 2c, left). In contrast, a second modality-specific group of MGB neurons was modulated by the trial epoch in a sensory modality-specific manner (Fig. 2c, middle). These results demonstrate that reward learning drives heterogeneous neuronal plasticity in MGB that flexibly reflects task features and reward outcomes in a modality-driven as well as modality-independent manner.

### Functional plasticity in MGB is not exclusively driven by behavioral variables
During the cross-modal Go/Nogo paradigm, mice adapted their behavior flexibly and reversibly. Behavioral variables such as body movement, licking[9,18,19] or arousal level[20] affect neural activity in a brain-wide manner. Depending on the trial epoch, ~5.7–38.6% of MGB neurons exhibited correlations of $Ca^{2+}$ activity with behavioral variables (Supplementary Figs. 6a, b, threshold: $|r| = 0.2$). However, the behavioral modulation of $Ca^{2+}$ activity was not systematically changed across learning (Supplementary Figs. 6c, d). In addition, we

found that the strength of the correlation of $Ca^{2+}$ activity and locomotion or pupil size in neurons that were tracked across the behavioral paradigm was unchanged or even decreased after learning and upon behavioral adaptation (Supplementary Figs. 6e, f). In addition to locomotion and arousal, the number of licks during the pre-reward delay period increased after learning (Supplementary Figs. 6g, h). Nevertheless, anticipatory licking did not systematically modulate MGB activity during expert phases (Supplementary Figs. 6i, j) and did not correlate with the proportion of ramping cells during the delay period (Supplementary Fig. 6k). Taken together, these results indicate that subsets of individual MGB neurons correlate with behavioral variables such as locomotion, pupil size and licking, yet changes in behavior are not the main driver of functional plasticity and changes in learning-related activity in MGB.

### Learning induces coherent neuronal population representations during reward-preceding periods
In addition to the classification of individual stable and plastic MGB neurons, we next asked if and how the neural population representation of MGB changes upon learning. To answer this question, we calculated the trial-by-trial population vector correlation (PVC, Pearsons's r)[21–23] from all recorded neurons of one session, including longitudinally tracked and non-tracked cells. Throughout an individual session as well as across learning, the PVC of the stimulus period (Go and Nogo trials) remained stable (Figs. 3a, b, and Supplementary Figs. 7 and 8). In contrast, the PVC specifically increased during the reward-preceding delay period of Go trials once mice learned the stimulus-reward contingency in initial learning and flexibly adapted to the previous non-rewarded delay period after reversal learning (Figs. 3c, d, Supplementary Figs. 7 and 8). Thus, upon associative learning, the neuronal representation of the MGB population becomes more similar during the reward-preceding delay period in a sensory modality-independent fashion (Supplementary Figs. 7 and 8). The PVC increase was specific to the outcome and not the action given that it occurred only during the delay period of Hit, but not False Alarm trials, where mice incorrectly licked to a Nogo stimulus (Figs. 3e, f, and Supplementary Figs. 9 and 10). Furthermore, the delay period PVC increase was present in trials with and without anticipatory licking (Fig. 3g) indicating that enhanced PVCs during the delay period are not driven by the preparatory behavior of the animal or lick impulsivity. Removing delay cells (i.e., Ramp-up and Ramp-down cells, see Figs. 2a, b; see Supplementary Fig. 11 for the removal of unspecific subclusters) did not affect the increase in PVC (Fig. 3h), indicating that the change in population representation during the delay period is driven by the total population of MGB neurons.

### Associative reward learning changes the co-activity network structure in MGB
Next, we analyzed how the co-activity network structure changes across learning in MGB using longitudinally tracked cells by computing the weighted undirected graphs of MGB population activity[24,25]. Here, nodes in the co-activity network represent individual MGB neurons and edges the positive pairwise Pearson's correlation coefficients of neural activity between all neuron pairs (Fig. 4a). During the delay periods of the Go trials, the hubness (average of the activity correlations between all cell pairs) increased from the naive to the expert phases, while it was unaffected in the Nogo trials (Fig. 4b). Furthermore, the mean shortest path length between any two neurons became shorter in the expert phase in the Go trials, which is consistent with a global increase of hubness (Fig. 4b). Local co-activity structures represented by the cluster coefficient of triad did not increase from the naive to expert phases (Supplementary Fig. 12a), indicating that the observed changes are a global event in the total MGB population. Upon reversal learning, the co-activity structure during the delay period was

**Single cell plasticity across learning**

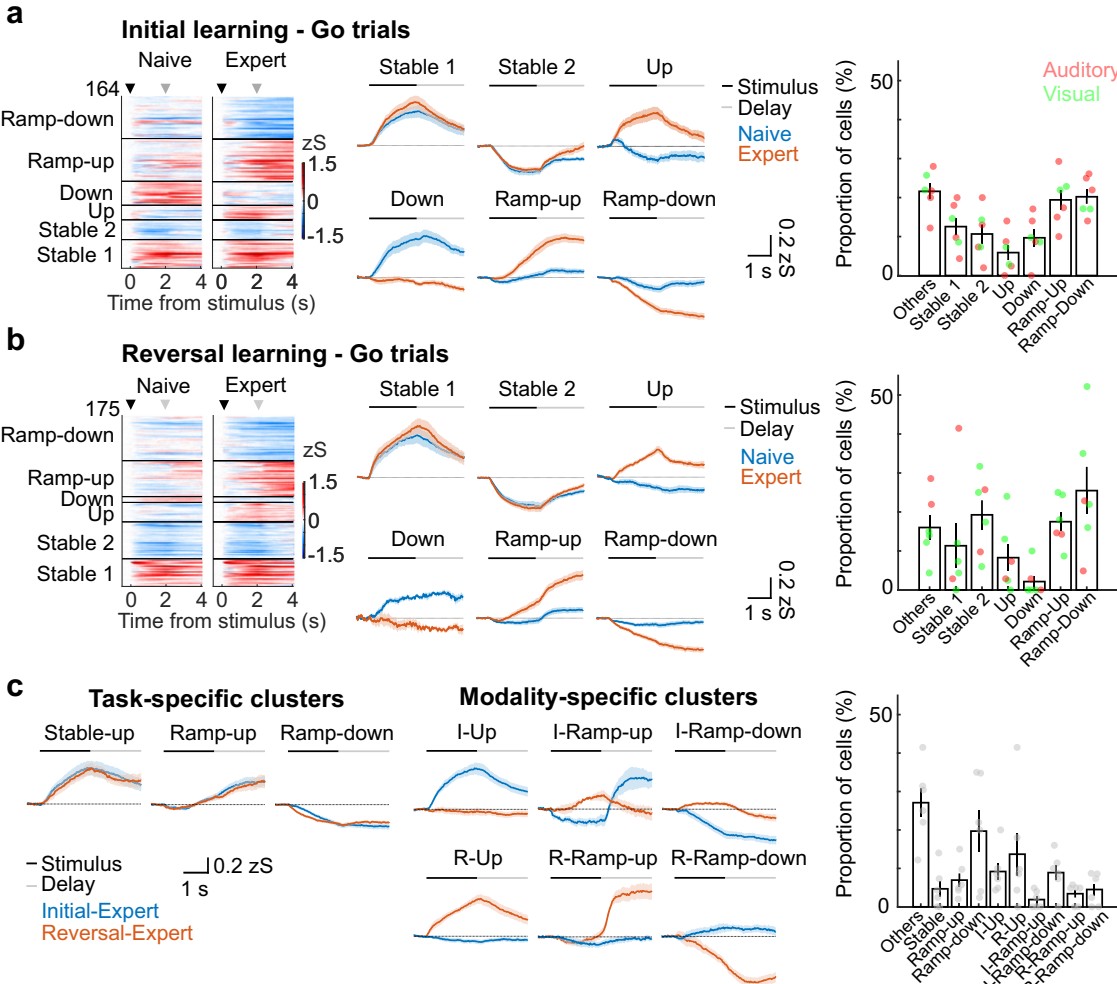

**Fig. 2 | Reward learning induces heterogeneous single cell plasticity in a task- and modality-specific fashion. a** Functional subgroups of MGB neurons in initial learning. Left: Heatmaps of single-cell activities before and after learning in initial learning. Cells were clustered into functional subgroups depending on their activity patterns (*n* = 164 cells from 6 mice). Middle: Average calcium traces (mean ± SEM) of the functional subgroups shown on the left side. Right: Proportion of cells in each cluster. Each dot represents the data from the individual mouse. Red and green dots represent the type of stimulus-reward association. **b** Learning-related functional subgroups in reversal learning (*N* = 6 mice, *n* = 175 cells). Figure structures are the same as (**a**). **c** Functional subgroups exhibiting task and modality-specific plasticity in the two expert phases in initial and reversal learning (*N* = 6 mice, *n* = 154 cells). Left: Average calcium traces (mean ± SEM) of the task-specific functional subgroups. Middle: Average calcium traces (mean ± SEM) of the modality-specific functional subgroups. Right: Proportion of cells in each cluster.

actively remodeled, both globally (hubness and path length) and locally (cluster coefficient) (Fig. 4b, Supplementary Fig. 12a), indicating that MGB co-activity network structure is dynamic during changing stimulus-reward contingencies. Furthermore, removing the cluster of delay ramping neurons (or a similar number of random neurons) from the analysis did not affect the changes in the co-activity network structure (Supplementary Figs. 12b, c). These results indicated that the general MGB population mediates the remodeling of the co-activity network. Anti-correlation networks (negative Pearson's r) remained unchanged in all learning stages (Supplementary Fig. 13). In summary, global co-activity network structures in MGB changed dynamically during the reward-preceding delay period and flexibly re-adjusted after learning in a modality-unspecific fashion. Changes in co-activity structures during the stimulus period were less consistent and limited to the reversal learning period (Supplementary Fig. 14). Altogether, these results demonstrate that the functional co-activity structure in MGB can be rapidly re-organized through associative reward learning and learning rule switches, which could help to support the cognitive processing of task-relevant information[24,25].

## Discussion

Recent work has shed light on the role of sensory thalamus in adaptive behaviors[3,4], while the neural dynamics of sensory thalamus in flexible learning of complex tasks across multiple sensory modalities remained elusive. Here, we combined a cross-modal (auditory and visual) reward-associative learning paradigm with longitudinal deep-brain two-photon calcium imaging in medial geniculate body (MGB) and demonstrated that sensory thalamus dynamically encodes sensory as well as task-related information, adapting its neural responses to varying task rules.

We find that auditory thalamus exhibits adaptive processing of auditory and visual information (Figs. 1i−m and Supplementary Figs. 15 and 16) similar to sensory cortices[15,16]. Responses of MGB neurons to auditory and visual stimuli are plastic and modulated upon cross-modal stimulus-reward learning (Fig. 1l, m), indicating that sensory encoding in MGB is dynamic upon cognitively demanding tasks irrespective of the sensory modality (Supplementary Fig. 3). A subset of MGB neurons developed ramping activity during the reward-preceding delay period (Fig. 2a−c), during which the animal has to hold

**Trial-by-trial population vector correlation**

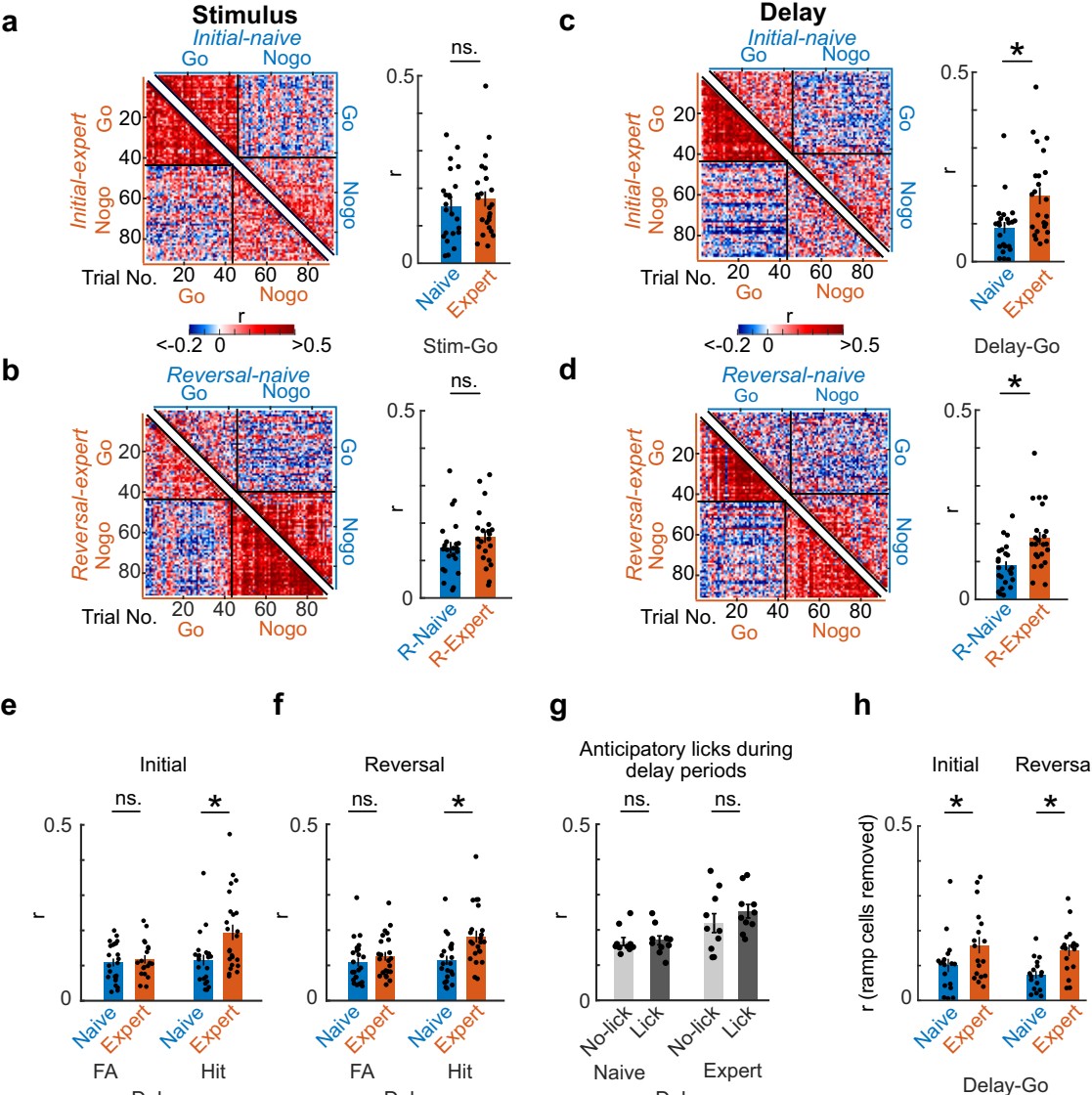

**Fig. 3 | Learning-induced remodeling of neural population level representations of task features in MGB. a–d** Representative single session trial-by-trial population vector correlation (PVC) matrices of stimulus (**a**, **b**) and delay (**c**, **d**) periods from one example mouse. Bar charts show the means of Go trial PVCs between naive and expert phases in initial (**a**, **c**) and reversal learning (**b**, **d**) from all mice (*N* = 8 mice, 2–3 sessions / mouse, see Supplementary Figs. 7–10). **e**, **f** Mean trial-by-trial population vectors correlation of the delay period of Hit and False Alarm (FA) trials in different learning stages (*N* = 8 mice, 1–3 sessions / mouse, see Supplementary Fig. 9). **g**, Mean trial-by-trial correlation in the delay period in Hit

trials with or without anticipatory licking in the initial learning (*N* = 8 mice, 1–3 sessions / mouse, see Supplementary Figs. 7 and 8). Due to the small number of the Hit trials with anticipatory licking in reversal learning, we excluded the reversal learning data for this analysis. **h** Mean trial-by-trial correlation in the delay period after removing all ramping cells found in Fig. 2a, b in all mice (*N* = 6 mice, 2–3 sessions / mouse, see Supplementary Fig. 11). Each dot is the mean *r* value in individual sessions in each training stage for all mice. Statistical tests: Linear mix model (**a**–**f**, **h**) *\*p* < 0.01, details in Supplementary Table 1). 2-side Wilcoxon rank sum test (**g**). Error bars: mean ± SEM (**a**–**h**).

the predictive information of the stimulus about the trial outcome, which could reflect reward anticipation[11] or short-term memory to guide the next action[26], similar to cortical and thalamic areas of the mouse motor system[27]. Reciprocal loops between motor thalamus and cortex have been shown to be necessary to maintain ramping activities in either region during delay periods[27], suggesting that, similar to the motor system, auditory thalamus could cooperatively maintain delay ramping activity together with auditory and other cortices.

On the population level, learning can either increase[28,29] or decrease[30,31] trial-by-trial correlation, whilst other cognitive factors such as attention[32,33] exhibit a bi-directional influence. In our reversal

learning task, the neuronal representation of sensory stimuli in auditory thalamus was stable across learning, regardless of whether they were predictive of a reward (Go cue) or not (Nogo cue). These findings are similar to previous observations for conditioned stimulus presentations in aversive auditory fear learning[5]. In contrast to the stimulus period, the MGB population response (trial-by-trial correlation) became more coherent across trials during the non-sensory pre-reward delay period, once mice learned the task-reward rule, irrespective of the sensory modality (Figs. 3a–d, Supplementary Figs. 7 and 8). This trial-by-trial population vector correlation increase during the delay period was specific to Hit trials, but not False Alarm

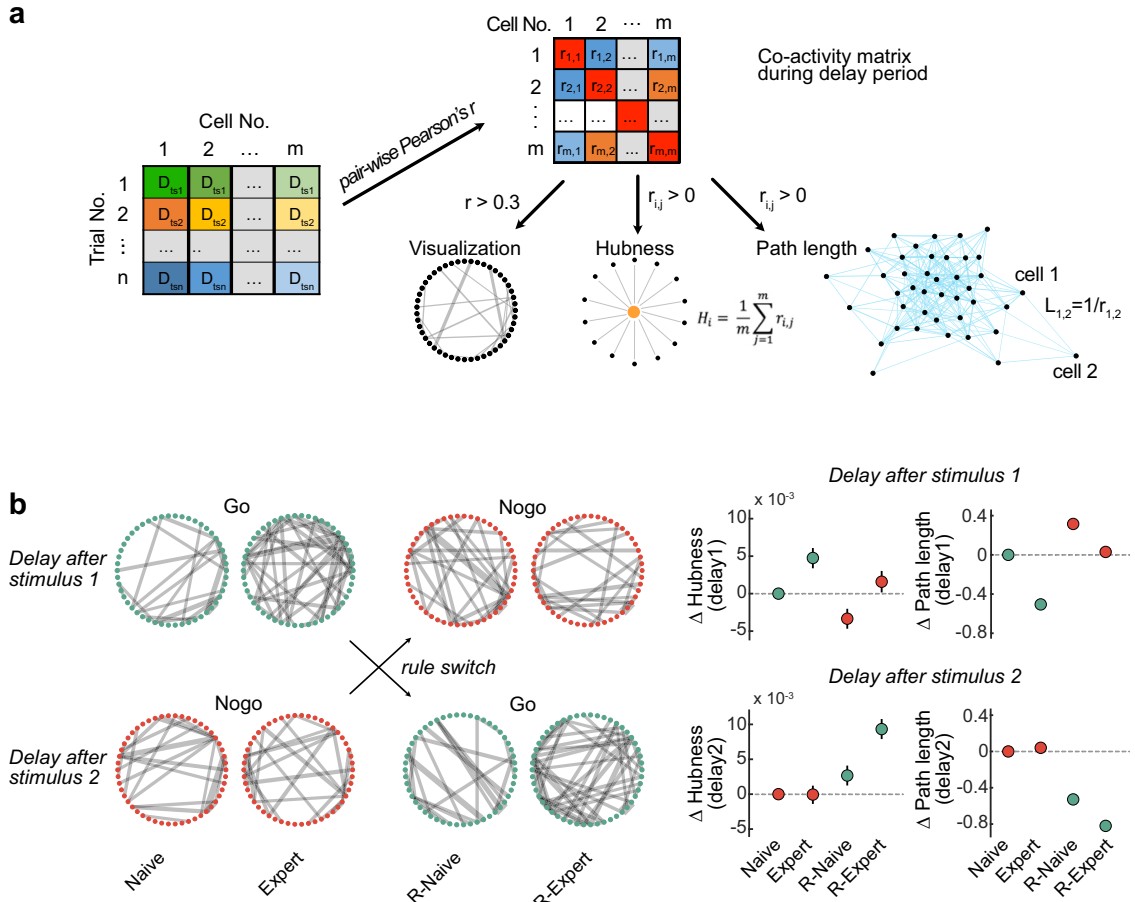

**Fig. 4 | Learning re-organizes MGB co-activity structure. a** Co-activity matrix construction and network analysis. The co-activity matrix represents the cell-by-cell pair-wise Pearson's correlation (r) from all concatenated time series vectors of the delay periods ($D_{ts}$) from all trials (see left matrix for construction of time series vector). The co-activity strength is visualized by line thickness in a circular plot for each cell pair ($r > 0.3$, dots indicate individual neurons). Global MGB co-activity strength is quantified by hubness and shortest path length. $D_{tsn}$: 2 s time series during delay period in trial n. $r_{i,j}$: Pearson's correlation of calcium activity between trials (Figs. 3e, f, Supplementary Figs. 9), indicating that the re-organization of the population activity during the delay period is dependent on the predicted task outcome and not the general reward preparation and consumption movement per se (e.g., licking, Fig. 3g). Recently, trial-by-trial correlations were suggested to account for the optimization of information communication between upstream sensory information encoding and downstream read-out guiding behaviors in different population representation subspaces[34–36]. Together, our findings link changes in population dynamics of auditory thalamus to a wider brain network that is implicated in functions of task representation[37], short term memory[27,38], outcome-dependent actions[39] and reward anticipation[11].

cell i and j. $H_i$: hubness of cell i to all its neighbors. $L_{1,2}$: path length between cell 1 and 2. **b,** Left: Representative circular plots show an example MGB co-activity network structure at different learning stages of Go and Nogo trials for one representative mouse before and after learning the rule switch. Right: Changes of hubness and path length across all neuron pairs in each learning stage which are baselined to the values in the naive Go (Stimulus 1) or Nogo (Stimulus 2) condition ($N = 210$ neurons from 6 mice, 3000 bootstraps). Error bars: 95 % confidence interval of mean.

Co-activity network structure during the reward-preceding delay period was remodeled across associative learning regardless of the sensory modality, while it remained by-and-large stable during the stimulus period (Fig. 4b, Supplementary Figs. 12 and 14). The distinct changes of co-activity network states between the stimulus and delay periods may result in differential information transfer to downstream areas to prime information integration and guide behavioral outputs in Go or Nogo trials[7,40,41]. While the global co-activity network changed upon learning (i.e., path length and hubness), the local co-activity network (i.e., cluster coefficient) did not systematically change during the delay period, and ramping cells did not contribute to the

enhancement of the co-activity network (Supplementary Fig. 12). This suggests that the activity of the total MGB population supports the functional network structure and not individual functional subgroups of neurons. Upon reversing the stimulus-reward contingency, the co-activity structures were dynamically remodeled not only during the delay but also during the stimulus periods (Supplementary Fig. 14), which indicates that reward-associative learning with changing stimulus contingencies might depend on a learning rule that results in the generalization of the Go stimulus from a sensory representation towards a predictive state representation that could be flexibly assigned to future learning rule updates. It remains to be tested if and how these changes are mediated and whether they are generated locally or if they are co-dependent on external activity, e.g., primary or associative cortical brain areas[42].

What are the neural mechanisms that guide functional plasticity in MGB from naive to expert phases across reward associative learning? Given the sparse local recurrent connectivity within mouse sensory thalamus[1,43,44] and negligible proportions of local inhibitory neurons[5,45], local microcircuit plasticity is unlikely to drive changes in response patterns. Plausible scenarios could include changes of synaptic plasticity, neuromodulation or adaptations of long range bottom-up excitatory[46,47] and/or inhibitory[3,4] inputs. In addition, corticothalamic feedback[2,48] could stabilize changes in MGB population activity upon

learning, either directly or di-synaptically via the thalamic reticular nucleus (TRN). Future studies will need to test how distinct circuit elements and their combination can affect functional population level plasticity in auditory thalamus as well as other thalamic nuclei.

While reward learning biased the sensory responses of subsets of MGB neurons towards reward-associated Go stimuli during the task regardless of sensory modality (Figs. 1l, m, Supplementary Fig. 3), the sensory responses in off-task non-rewarded mapping sessions before and after learning (i.e., passive measurements of tuning to multimodal sensory stimuli) were not biased to the on-task Go stimuli (Supplementary Figs. 15 and 16). Specific multisensory enhancement to the Go stimulus, which was previously observed during appetitive learning in MGB[10], did not take place during off-task, non-rewarded sessions (Supplementary Figs. 17 and 18). This could be due to a fast devaluation of reward-conditioned stimuli[11]. Finally, while uni- and multisensory responses could be altered on the single cell level (Supplementary Figs. 15 and 16), sensory stimuli could be reliably decoded from the MGB population activity across learning (Supplementary Fig. 19). The complementary mechanisms of single cell plasticity and population-level stability of sensory coding could be crucial to allow for dynamic neuronal representations upon learning, while ensuring stable representations of the environment[5,49] balancing relay and cognitive functions of MGB.

Increasing evidence driven by dense electrophysiological recording techniques and deep brain imaging of identified cell types in head-fixed or freely moving animals reveals diverse functions of individual neurons in different thalamic nuclei during learning and adaptive behavior. For example, neurons in paraventricular thalamus exhibit heterogeneous response adaptation during cue-reward learning and reward seeking, which are suppressed by fearful stimuli[50,51]. Similarly, neuronal responses in MGB are adaptive to reward- (see above) and aversive-outcome[5,12] predicting sensory stimuli during associative learning. Furthermore, sensory responses in higher order somatosensory and visual thalamus are modulated by attention and reward-predicting stimuli even by non-classical modalities similar to our findings[52,53]. Despite these response similarities, it is currently unclear if and how distinct thalamic areas including first order 'relay nuclei' as well as higher-order and non-sensory thalamus interact together with cortex and other brain areas to shape learning. Multisite large-scale recording approaches[54,55] and (all-optical) perturbations of information flow between thalamic subnuclei and their non-thalamic inputs and outputs might help to reveal the distributed population code during adaptive behavior.

Altogether, our study reveals that auditory thalamus displays flexible adaptations of single cell responses and co-activity network states that align not only with sensory but also task-period and outcome-relevant information, which change bi-directionally upon updated stimulus contingencies in reward-associative learning, highlighting the role of sensory thalamus in complex neural computations for adaptive behaviors as part of a wider network of thalamic nuclei for cognitive function.

## Methods

### Animals

All experiments were performed in accordance with the institutional guidelines of University of Basel or DZNE, Bonn and were approved by the Cantonal Veterinary Office of Basel-Stadt, Switzerland or the Landesamt für Natur, Umwelt und Verbraucherschutz Nordrhein-Westfalen, Germany, respectively. Six to 10-week-old (at the start of the experiment) male C57BL/6 J mice were used throughout this study. Animals were housed on a 12-h light / dark cycle at an ambient temperature (22 °C) and humidity (55 %) and had free access to food and water until the initiation of the behavioral experiment. Throughout the behavioral experiment, animals were placed under food restriction and their body weights were maintained at 85–90% of their free-

feeding weights. Well-being was monitored daily through the entire experimental period. No statistical methods were applied to pre-determine the sample size for each experiment. The investigator was not blinded for surgery, behavioral and imaging experiments or data analysis.

### Surgical procedures

Buprenorphine (0.1 mg/kg) was subcutaneously injected ~30 min before surgery for analgesia. Then, mice were anesthetized with iso-flurane (1.5–2.0 % maintenance) through the oxygen-enriched air (95 %, 1–3 l/min, Oxymat III, Weinmann). Anesthesia level was monitored via breathing rates and foot and tail reflexes before and during surgery. Mice were placed in a stereotaxic apparatus (Model 1900, Kopf Instruments), and their body temperature was maintained through a heating pad (Rodent warmer, 53800 M, Stoelting). Their eyes were covered by an eye protective cream (Bepanthen Augen und Nasen-salbe, Bayer). A mixture of Lidocaine (10 mg/kg) and Ropivacaine (3 mg/kg) was injected under the skin over the skull for local anesthesia. Stereotaxic viral injections were performed as previously described[5]. Briefly, a small craniotomy was performed above the medial geniculate body (MGB, AP: −3.28, ML: −1.9, DV: −3.0 mm) by using a stereotaxic drill (Model 1911, Kopf) with a burr drill bit (105-0135-225, Kyocera). A pulled glass pipette (2-000-001, Drummond Scientific) filled with AAV vector was slowly lowered into the brain with the help of a micropositioner (Model 2650, Kopf). AAV2/1.syn.jGCaMP7s[56] (Addgene, 104487-AAV1, ca 500 nl, diluted by sterile PBS, 1-2x) was injected into MGB with a pressure ejection system (Picospritzer, Parker). One to two weeks after viral injection, a gradient refractive index (GRIN) lens (0.6 mm diameter, 7.3 mm length or 1.0 mm diameter, 4 mm length, Inscopix) was implanted during the second surgery (anesthesia and analgesia, see above). 0.6 mm lenses were implanted as previously described[5]. Briefly, a 0.8 mm diameter craniotomy was performed above MGB (drill: 105-0709.400, Kyocera) and a small track was cut with a 0.7 mm sterile needle. Next, the GRIN lens was slowly advanced into the brain using the Micropositioner (Model 2650). For the implantation of 1.0 mm diameter lenses, a 1.2–1.3 mm craniotomy was performed above MGB using a hand drill (503599, World Precision Instrument) with a burr drill bit (200 μm diameter, C1.104.002, Bösch Dental). Tissue above MGB was slowly aspirated through a sterile blunt needle (27 G, Endo irrigation cannula) connected to a suction system. Sterilized phosphate buffer saline (PBS) was used to irrigate the brain until the bleeding stopped around the aspirated site. Next, the 1.0 mm lens was slowly advanced into the brain with the micropositioner. Both, 0.6 and 1.0 mm lenses were fixed to the skull with light curable glue (Loctite 4305, Henkel). A custom-made head bar was attached to the skull next to the GRIN lens, and the skull was sealed with Scotchbond (3 M), Vetbond (3 M) and dental acrylic (Paladur, Kulzer, Orth Jet, Lang Dental and/or C&B Super-Bond, Sun Medical). Meloxicam (5 mg/kg) was injected subcutaneously after the surgery for post-operative analgesia.

For optogenetic experiments, either AAV2/5-hsyn-Jaws-KGC-GFP-ER2[57] (Addgene, 65014-AAV5, ca 500 nl, diluted by sterile saline, 2x) or AAV2/5-hSyn-EGFP (Addgene, 50465-AAV5, ca 500 nl, diluted by sterile saline, 2x) was bilaterally injected into MGB (AP: −3.2, ML: ±2, DV: −3.0/−3.3 mm) using the same methods as described above. Following the AAV injection, optical fibers were bilaterally implanted above MGB (AP: −3.2, ML: ±2, DV: −3 mm) using the Micropositioner on the same surgery. Other surgical procedures including anesthesia and local analgesia were the same as for GRIN lens implantations. Systemic analgesia was provided via carprofen in the drinking water (0.067 mg/ml) from ca. 12–24 h pre-surgery to ca. 72 h post-surgery.

### Behavioral apparatus

The behavioral apparatus was housed in a light shield chamber under a custom-built two-photon microscope (Independent NeuroScience

Services (INSS), UK). Mice were head-fixed by a custom-designed holding system and placed on a running-wheel connected to a rotary encoder (E6A2-CS3E, Omron) to measure locomotion. Auditory and visual stimuli were presented with a speaker (ES1, Tucker Davis Technologies, placed at upper-right, 10 cm from the mouse head) and a 7-inch screen (Adafruit 1667, placed 10 cm from the right side of the mouse face at a parallel angle), respectively. The screen system was modified to minimize the light exposure to photomultiplier tubes (PMTs) during a visual stimulus presentation in the two-photon imaging[58,59]. During the experiment, the gray background was continuously presented from the screen. A lick spout was mounted on the custom-built retractable stage controlled by Trinamic motion control language (TMCL). A reward (soy milk) was delivered by a custom-designed peristaltic pump system by using a micropump (mp6, Bartels Mikrotechnik). A licking of a reward spout was detected by a lick detector modified from the detector described in a previous study[60]. The experiments were controlled by a custom-written program in MATLAB (MathWorks, Psychophysics Toolbox, http://psychtoolbox.org) with NI USB-6008 (National Instruments) and RZ 6 (Tucker Davis Technologies), and the timing of TTL input/output of behavioral events were recorded by RZ 6 at 50 kHz sampling rate.

## In vivo two-photon calcium imaging
In vivo two-photon calcium imaging was performed using a custom-built two-photon microscope (INSS, UK). The microscope was equipped with a resonant scanning system and a pulsed Ti:sapphire laser ($\lambda = 940$ nm, Chameleon Vision S, Coherent). A motorized three-axis system (Zaber motor) connected with a microscope head (Z-direction) and breadboard under the behavioral apparatus (X-Y directions) enabled locating an objective lens above the GRIN lens. The microscope system was controlled by ScanImage software (Vidrio Technologies). Green and red fluorescent photons were collected with an objective lens (x16, 0.80 NA, Nikon). Photons were separated by a dichroic mirror (T565lpxr, long pass, Chroma) and barrier filters (green: ET510/80 m, red: ER630/75 m), and measured by PMTs (PMT2101, Thorlab). The imaging frame was $512 \times 512$ pixels, and the frame rate was ~30 Hz. Fields of views (FOVs) of two-photon images were ~330 μm × 330 μm (at 3.0x zoom, 1.0 mm diameter GRIN lens, $N = 6$ mice) or ~400 μm × 400 μm (at 2.5x zoom, 0.6 mm diameter GRIN lens, $N = 2$ mice). Note that GRIN lens FOVs do not correspond to the actual size of the imaged brain areas due to the spatially non-uniform optical distortion inherent to GRIN lenses[61].

## Two-photon Image Processing
Two-photon images were processed using Suite2P[62]. The images were motion-corrected, and regions of interest (ROIs) were automatically generated. Next, experimenters curated ROIs and sorted them as neurons or not. A small portion of ROIs were drawn using a manual ROI drawing function of Suite2p. A neuropil signal was also calculated for each ROI by Suite2P. A correction coefficient (0.7) was multiplied with the neuropil signal, and each ROI signal was subtracted from this value and handled as $Ca^{2+}$ signal of individual neurons. For behavioral sessions, we selected 2–3 sessions per learning phase (initial naive, initial expert, reversal naive and reversal expert) for each mouse. The naive phase consisted of the first 2–3 behavioral sessions, and the expert phase consisted of the 2–3 sessions in which d-prime (d′, discriminability index)[63] reached over 1.5. Across these selected sessions, we tracked the same ROIs wherever possible. The ROI tracking procedure was done separately for behavioral and sensory mapping sessions. The same ROI signals, or matched cell data, were used for the following analyses: k-means clustering of the Go/Nogo task (Figs. 2a–c, Supplementary Fig. 4), co-activity network structure analysis (Fig. 4 and Supplementary Figs. 12–14), correlation analysis for behavioral variables and $Ca^{2+}$ signals (Supplementary Figs. 6a–f) as well as the sensory mapping analysis (Supplementary Figs. 15–19) unless stated otherwise.

For the trial-by-trial population vector correlation analysis, co-activity network structure analysis as well as decoder training (see below), $Ca^{2+}$ signals of individual neurons were detrended and lowpass filtered to 5 Hz with a Butterworth filter.

## Longitudinal cell tracking and signal extraction
Two-photon images acquired in the first imaging session were used as template. In the following sessions, we returned to this template image plane at the beginning of two-photon imaging. To return to the same imaging plane, several cells with bright signals and clear contours were used as reference cells. A custom, tightly fit head bar and holder design prevents angular rotation of the image plane aiding the reliably identification of the same imaging plane to track as many individual cells as possible. To extract calcium signals from the tracked cells across imaging sessions, "RoiMatchPub" (https://github.com/ransona/ROIMatchPub) was used as described above ("Two-photon Image Processing"). RoiMatchPub matches the data of regions of interest (ROI) across sessions based on Matlab files generated by Suite2p (https://github.com/MouseLand/suite2p; see also https://suite2p.readthedocs.io/en/latest/multiday.html). RoiMatchPub uses a semi-automatic cell tracking algorithm. At first, a template two-photon image was prepared, i.e., the max-intensity projection (MIP) two-photon image acquired in the first imaging session, as well as the MIP images of the following sessions. Then, ROIs were manually matched in the General User Interface of RoiMatchPub between the first template two-photon image and two-photo images in following sessions (e.g., 1st session image – 2nd session image, 1st session image – 3rd session image, 1st session image and 4th session image and so on).

## Sensory mapping
Before the start of the sensory mapping sessions, mice were head-fixed under the custom-built two-photon microscope and habituated to the behavioral apparatus and environment for minimum 3 days. Each session started with a 1 min habituation period. Auditory (4, 8, 12, 16 or 20 kHz pure tones at 75 dB, 2 s), visual (Upward, downward, rightward or leftward sine wave drifting gratings, 100% contrast, 2 Hz, 0.05 cycle per degree, 2 s) or multisensory stimuli (combination of auditory and visual stimuli, e.g., 4 kHz pure tone with upward drifting grating, 2 s) were presented. The sensory stimuli were presented in a pseudo-random order. Each auditory, visual and the multisensory stimulus was presented 8 times (e.g., 4 kHz pure tone, rightward grating, and the combination of them were presented for 8 times), and a total of 240 trials (8 trials x 30 stimulus types of uni and multisensory stimuli) were performed in one session per day. Inter-trial interval (ITI) was 6-9 s. For one mouse, each stimulus was presented 5 times and a total of 150 trials / session were performed. Sensory mapping was performed for 2-3 consecutive sessions before and after the sensory Go/Nogo reversal learning paradigm.

## Behavioral training in a sensory Go/Nogo reversal learning task
Following sensory mapping sessions, mice were pre-trained to lick a spout to receive a liquid reward (soy milk) for 1–2 days under head-fixation. Next, the animals were trained to perform a sensory Go/Nogo task, which consisted of Go trials (30%), Nogo trials (35%) and catch trials (35%) (total number of trials: 140 per session). At the beginning of each trial, a 6–13 s ITI was initiated. Then, either an auditory stimulus (Go cue, 12 kHz pure tone, 75 dB, 2 s), a visual stimulus (Nogo cue, rightward drifting grating, 2 s) or no stimulus (2 s blank period, catch trial) was presented. The sensory stimulus or blank period was followed by a delay period (2 s) without any sensory stimulus. At the end of the delay period, a response window (1.5 s) was initiated and a retractable lick spout moved forward to the mouse. Go trials required mice to lick the spout (Hit) to obtain a reward (8-10 μl soya milk), otherwise the trial was considered as an error (Miss). In Nogo trials, mice were required to withhold the lick response (Correct Rejection, CR). If the mice licked the

spout in the Nogo trial, the trial was considered as an error (False Alarm, FA). In catch trials mice were required to withhold the lick response (catch-CR). If mice licked the spout, the trial was considered as an error (catch-FA). If mice accidentally touched the spout too early (within 200 ms after the response window onset) (e.g., due to grooming), the contact was not considered as a response. Catch trials were introduced to ensure that mice identify the Go cue as a reward predictor. Task performances for Nogo and catch trials developed similarly (Figs. 1d, e, and Supplementary Fig. 1). Thus, the task performances in those trials were combined to calculate the task performance index, d-prime (d') as follows: d' = Z (Hit ratio) - Z (False alarm ratio). Hit and False alarm ratios were calculated as follows: Hit ratio = the number of Hit trials/(the number of Hit trials + the number of Miss trials); False alarm ratio = the number of false alarm trials/(the number of false alarm trials + the number of correction rejection trials). If the ratio reached 1.0, the ratio was adjusted to calculate d'[63]. Z is the inverse cumulative distribution function, and Z (Hit ratio) and Z (False alarm ratio) were calculated by using the qnorm function (https://de.mathworks.com/matlabcentral/fileexchange/48978-qnorm-matprobabilities-dblmean-matsigmas-bool useapproximation). Once d' reached values above 1.5 for three consecutive days, mice were considered experts and a reversal learning paradigm was initiated. Upon reversal learning, the stimulus-reward contingency was switched. In the group in which the auditory stimulus was presented as a Go cue, the visual stimulus was now presented as a Go cue and rewarded and vice versa for animals in which the visual stimulus was initially presented as a Go cue. Once d' reached values above 1.5 for three consecutive days, mice were considered reversal experts. The order of the stimulus-reward contingency was counter-balanced between mice, i.e., four mice were trained to the auditory stimulus as a Go cue first and four mice were trained to the visual stimulus as a Go cue first. Specific cases of data exclusion: Mice were trained for a maximum of 28 sessions. Due to slower learning, one out of eight mice reached only two reversal expert sessions before the training had to be terminated. Furthermore, in two out of eight mice the training had to be briefly suspended due to technical issues, which did not have a major impact on task performance after re-initiation of the paradigm. In one mouse, an imaging artifact gradually appeared from the lateral edges of the two-photon image in one session. Data from the later stage of this session was excluded from the analysis. In one mouse, the data recording of one session was aborted in the middle of the session due to the malfunction of the acquisition system. Since the number of trials reached more than half of the session, the data obtained in this session was included for the data analyzes.

### Optogentic inhibition
Behavioral training procedures were the same as mentioned. The auditory stimulus was presented as a Go cue, while the visual stimulus was presented as a Nogo cue. No stimulus was presented in catch trials. Optogenetic inhibition was applied to all Go, Nogo and Catch trials during a stimulus presentation period for 2 s with a 250 ms ramp-down period at the end of optogenetic inhibition to avoid rebound activity after inhibition[57]. 9 mW (at fiber tip) of 617 nm light (Thorlabs, M617F2) was bilaterally delivered to MGB via a 400 μm optical fibers (Thorlabs, M98L01) and cannulas on each side (DORIC, multimode, 0.37 NA).

### Video recording
Mouse behavior was monitored during the experiment with a CMOS camera (a2A2590-60umPRO, Basler) equipped with a CCTV lens (Moritex ML-M1616UR) and a band-pass filter (DB850, Midwest) ($N = 7$ mice) or a Raspberry Pi Camera Module 2 with a shortpass filter (FES850, Thorlabs) controlled by Raspberry Pi 3 model B+ ($N = 1$ mouse). Either camera system was located on the left side of the mouse together with a custom-made infrared LED system (830 nm). In the CMOS camera system, each frame acquisition was synchronized with the acquisition of a two-photon image (ca. 30 Hz) through ScanImage.

In the raspberry Pi camera system, each frame was acquired at ca. 30 Hz in free-run mode and image acquisition was monitored by internally-generated TTLs. During offline analysis, the tongue and pupil were detected and tracked by animal pose estimation using DeepLabCut (DLC)[64]. The experimenter manually labeled the tongue and the eight points on the edges of the pupil (top, top-right, right, bottom-right, bottom, bottom-left, left, top-left) for each mouse to train the model. Videography based licking-behavior was detected if the tongue was tracked by DLC for at least two consecutive frames. The pupil area was calculated by using a circle fit function[65] (fitcircle, https://mathworks.com/matlabcentral/fileexchange/15060-fitcircle-m?s_cid=ME_prod_FX) from available data points at each frame. Pupil area data was smoothed by using a Hampel filter (MATLAB built-in function, number of neighbors, 10; number of standard deviations, 1.0).

### Mean individual cell activity during Go and Nogo trials
Heatmaps were used to visualize mean activities of individual MGB neurons pooled from all mice across learning ($N = 8$ mice, Fig. 1i, j). For the naive phase, the data of the first training session in both initial and reversal learning were used to show the unconditioned responses to sensory stimuli. For the expert phase, the data from the session with the highest d' in both initial and reversal learning were used to show well-conditioned responses to the sensory stimuli. The calcium data was baselined to the mean during 0.5 s before stimulus presentation in each trial. The data of individual calcium traces were averaged across Go and Nogo trials separately (maximum 42 trials for Go trials, and maximum 49 trials for Nogo trials in a single session). Cell IDs were sorted according to the amplitude of the mean sensory response during the stimulus presentation (2 s) in the Go trials.

### Proportion of stimulus-responsive cells in the sensory Go/Nogo task
Stimulus-responsiveness was determined through a two-step procedure. First, we performed a signed-rank test to examine if the sensory response of each cell was significantly different from zero. In each cell, the sensory response during the stimulus presentation (2 s) was averaged in Go and Nogo trials. Then, the means of the sensory responses pooled across Go or Nogo trials were used for a signed-rank test of each cell. Cells with statistical significance in the signed-rank test were selected for response thresholding (threshold: median z-score > ± 0.2). In addition, for auditory stimulus trials, the sensory response during the stimulus onset (0.3 s) was averaged and analyzed in the same manner as described above to catch fast-adapting MGB neurons. If a neuron was classified as sensory responsive during the whole 2 s stimulus presentation period or as onset responsive, the neuron was included as auditory responsive. Neurons where then classified as Go, Nogo or 'both' responsive cells depending on the trial type. Neurons that did not pass the detection threshold were classified as non-responsive cells.

### K-means cluster analysis
K-means cluster analysis was performed to sort individual neurons into functional subgroups. The calcium traces of matched individual neurons ($N = 6$ mice, $n = 210$ cells) were averaged across Go or Nogo trials for 2–3 sessions in each training phase (naive, expert, reversal naive and reversal expert). The mean Go or Nogo responses of each neuron were concatenated between naive and expert phases in both initial and reversal learning (time-series concatenation). The time-series for each cell was composed of the concatenated stimulus and delay periods of native and expert training phases, while initial and reversal learning were treated as independent observations. This times series underwent principal component analysis followed by k-means clustering (cosine distance) to sort the individual neurons into functional clusters. Thirty clusters were generated, and clusters showing similar

activity patterns were merged manually[5]. After generating the merged clusters, cell IDs in each merged cluster were separated to the initial and reversal learning data. To track the activity patterns of the MGB neurons between the two expert phases in initial and reversal learning, we performed k-means cluster analysis by using the data of the two expert phases in the initial and reversal learning (Fig. 2c, Supplementary Figs. 4a–c). After the preprocessing described above, the data of the two expert phases were concatenated between initial and reversal learning, then k-means clustering was performed.

### Correlation analysis between behavioral variables and neuronal activity

Correlations between behavioral variables (locomotion, pupil size) and single cell Ca²⁺ traces were calculated as the Pearson's correlation coefficient in Hit trials across learning (Supplementary Figs. 6a-f). The first session in the naive phase and the session with the highest d-prime in the expert phase were selected in both initial and reversal learning. Pupil, locomotion and calcium data of day-matched cells was downsampled to 5 Hz. Cells that exhibited correlations between the behavioral variable and the calcium data of $|r| > 0.2$ and $p < 0.05$ were considered as significantly correlated. R-value distributions were compared between the naive and expert phase in both initial and reversal learning by a Kolmogorov-Smirnov test (ks-test2, MATLAB). To measure the change of correlation across learning, the difference of $|r|$ values between naive and expert phases in matched cells was measured for stimulus, delay and ITI periods (signed-rank test). The pupil data was analyzed in the same manner except that the pupil data was smoothed by a Hampel filter (see above) and z-scored (Supplementary Figs. 6b, d, f).

### Anticipatory licking analysis

Anticipatory licking was quantified during the delay period. To visualize how the number of anticipatory licks change across learning, the proportion of trials with anticipatory licking was plotted as a function of the number of anticipatory licks per trial. The number of trials with or without anticipatory licking was pooled from all mice in each training phase, and this pooled data was used for chi-square testing (Supplementary Fig. 6g). The probability of anticipatory licks per time bin was calculated across mice (Supplementary Fig. 6h). For visualization purposes, lick probability was smoothed by a moving average (5 frames, movmean, MATLAB). To examine how anticipatory licking influenced MGB activity during the delay period, Ca²⁺ traces of Hit trials with and without anticipatory licking were separated and baselined to the mean Ca²⁺ fluorescence 150 ms before the delay period onset (Supplementary Fig. 6i, same baseline duration for Supplementary Fig. 6j).

### Trial-by-trial population vector correlation

Trial-by-trial population vector correlation (PVC, Pearson's r) was analyzed in naive (2–3 sessions) and expert stages (2–3 sessions) in both initial and reversal learning. The definition of the naive and expert stages is described above ("Two-Photon Image Processing"). Some mice learned the task quickly and d-prime exceeded 1.5 in the third behavioral session in the naive stage. In this case, two sessions were included in the analysis of the naive stages. One mouse reached only two reversal expert sessions before the training had to be terminated. Before the construction of the population vector, calcium data for each cell was baselined to the mean of the 2 s pre-stimulus period on a trial-by-trial basis. Next, the mean calcium responses during the stimulus and delay periods were calculated and the population vector constructed. PVCs were calculated across the stimulus and delay periods of all trials and plotted as a N-by-N correlation matrix per session ($N$ = number of trials). Mean correlation values from all trial pairs in each session were used for summary statistics. A linear mixed model was used to compare the difference of R values (fixed effect: naive vs

expert, random effect: mouse ID). For a fair comparison between trials with and without anticipatory licking, only sessions with more than 6 trials with and without anticipatory licking each were included in the analysis (Fig. 3g). Reversal learning was excluded from the analysis of anticipatory licking due to low trial numbers. To examine the contribution of ramping cells to the PVC during the Go delay period ramp-up and ramp-down cells were removed from the PV, and PVC values of the Go delay period were re-calculated. The same number of cells as ramp-up and ramp-down cells were randomly removed to generate the shuffle dataset (nShuffle = 30, Supplementary Fig. 11).

### Weighted graph-based analysis of MGB co-activity network structure

Undirected weighted graph that represents the relationship of calcium activity between MGB neurons were computed based on matched cell data ($n$ = 210 neurons). In the graph, nodes and edges stand for cell identities and the pairwise Pearson's correlation coefficients between two neuron pairs, respectively. The calcium activity data was baselined to the mean of the 2 s pre-stimulus period of each trial. The pair-wise correlation coefficient of the time-series data (2 s) during the stimulus and delay periods was calculated between all cell pairs in Go and Nogo trials. Each graph with the number of nodes M, which are identical to the number of matched cells is represented by its adjacency M-by-M symmetric matrix C where each element $r_{ij}$ is the Pearson's correlation coefficients ($-1 \leq r \leq 1$) between two nodes (neurons) i and j. Positive and negative edges were analyzed separately (Supplementary Fig. 13). The strength of the global co-activity network structure among MGB neurons was quantified by hubness, i.e., the mean of the pair-wise activity correlation of each node to all the others:

$$H_i = \frac{1}{m} \sum_{j=1}^{m} r_{i,j} \tag{1}$$

We estimated the global communication efficiency between any two nodes by measuring the geodesic (shortest) path length in the graph. First, a weighted graph with the length between two nodes, i and j as $L_{ij} = 1/r_{ij}$ was created. Next, the shortest path length between any two nodes was computed using Dijkstra's algorithm among 3903 cell pairs. The local co-activity network structures were quantified by the cluster coefficient of the triad. The cluster coefficient of the triad was calculated among 50376 triads[66]. $w_{ij}$ is the edge weight between any two nodes. $\hat{w}_{ij}$ is the normalized edge weight by the maximum weight in the adjacency matrix between cell i and j:

$$C_i = \frac{1}{k_i(k_i - 1)} \sum_{j,k} (\hat{w}_{ij} \hat{w}_{ik} \hat{w}_{jk})^{1/3} \tag{2}$$

$$\hat{w}_{ij} = w_{ij} / \max(w)$$

The same number of cells as ramp-up and ramp-down cells were randomly removed to generate the shuffle dataset (nShuffle = 30, Supplementary Fig. 12). Network structure was recalculated using the shuffle dataset. Functional connectivity parameters were calculated using the BrainConnectivity toolbox (https://github.com/jblocher/matlab-network-utilities/tree/master/BrainConnectivity)[67]. Plotting and confidence interval calculation were adapted from DABEST toolbox (https://github.com/ACCLAB/DABEST-Matlab)[68].

### General sensory responsiveness

To quantify the general sensory responsiveness of MGB neurons before and after the Go/Nogo learning in sensory mapping session (Supplementary Fig. 15), the calcium data during the stimulus presentation (2 s) was split in half (0–1 s and 1–2 s). The mean of the calcium data during these two periods was handled as individual data

points. Next, it was tested whether individual MGB neurons were responsive to auditory, visual and/or multi-sensory stimulus through the same two-step procedures as described above. In multi-sensory trials, the data from all visual grating directions were averaged, and the sensory response to auditory frequencies was tested. In Supplementary Fig. 15c, cells were defined as excited/inhibited if they were responsive to any auditory frequency or grating direction. In Supplementary Fig. 15d, the response amplitude of the cells that were responsive to at least one of the sensory stimuli in pre-learning was compared to the response amplitude of the same sensory stimuli in post-learning sessions. In Supplementary Fig. 15e, change of peak tuning frequency or grating direction was calculated if a cell was responsive to any frequency/direction in both pre- and post-learning session. In Supplementary Figs. 15g, h, the time series of the averaged population response and the response amplitude of the cells with the significant response to the reward-associated stimulus in Go/Nogo training (12 kHz, rightward drifting grating, 12 kHz with rightward drifting grating) were compared between the pre- and post-learning session. In Supplementary Fig. 15i, the change of peak tuning frequency or grating direction was calculated if a cell was responsive to the conditioned stimulus (12 kHz, rightward-drifting grating, 12 kHz with rightward drifting grating) both in pre- and post-learning session.

## K-means clustering of tuning curves

To sort MGB neurons into groups with similar frequency tuning patterns, k-mean clustering ('correlation' distance) was performed with 20 features (5 frequencies, multi-/uni-sensory and pre/post training) (Supplementary Fig. 17). Sensory responses to the auditory (4, 8, 12, 16 and 20 kHz) and multi-sensory stimuli (4, 8, 12, 16 and 20 kHz with drifting gratings) were averaged across trials and sessions (1-3 sessions). Visual stimulus feature (i.e., grating directions) was collapsed in multi-sensory trials. The calcium data in the response period was baselined to 0.5 s before stimulus presentation. The mean of the calcium activity during the stimulus presentation (2 s) was used to generate frequency tuning curves. F: mean calcium response during stimulus presentation period. m: cell number (pooled across 6 mice). f: auditory frequency. Clustering matrix:

sensory index calculation. Cells with different calcium response sign in multi-sensory response and sum of uni-sensory response were excluded ($A_f V \cdot (A_f + V) < 0$). Absolute value of sensory response was taken to demonstrate the principle of multi-sensory integration (linear or non-linear). Visual grating directions were averaged in multi-sensory response as well as uni-sensory response. First order exponential fitting was conducted to reveal the trend of data distribution for each auditory frequency (Supplementary Fig. 17d). Multi-sensory indexes of all individual cells and all auditory frequencies were compared regardless of response sign and amplitude (Supplementary Fig. 18). Two-dimensional two-sample ks-test was performed to test if the distribution of the multi-sensory index was comparable between pre- and post-learning (See, Supplementary Table 1).

$$Multisensory\ index_f = \frac{\left| A_f V \right|}{\left| A_f + V \right|} \quad f = 4, 8, 12, 16, 20\ kHz \qquad (3)$$

## Single cell correlation analysis across multisensory mapping sessions

Single cell averaged peri-stimulus time histograms (PSTH) of uni- and multisensory trials for stimulus features (auditory frequency, grating direction) in pre- and post-learning were sorted by stimulus amplitude (0–2 s) and plotted as heat maps (Supplementary Fig. 16). Next, the Pearson's correlation between the response time series of individual neurons before and after learning was calculated and average across all cells (Supplementary Fig. 16)[69].

## Decoder analyses

Linear support vector machine (SVM) decoders were trained on the stimulus period (normalized by 1 s pre-stimulus baseline) for all tracked neurons in each multi-sensory mapping session for each animal. Time series data were down-sampled to 3 Hz to avoid overfitting. The decoders trained on individual sessions were tested in a pair-wise manner for all sessions (Supplementary Fig. 19a). For modality decoding (Supplementary Fig. 19b), decoders were

$$\begin{pmatrix} ^{pre}_{multi}F^1_{f1} & \cdots & ^{pre}_{multi}F^1_{f5} & ^{post}_{multi}F^1_{f1} & \cdots & ^{post}_{multi}F^1_{f5} & ^{pre}_{uni}F^1_{f1} & \cdots & ^{pre}_{uni}F^1_{f5} & ^{post}_{uni}F^1_{f1} & \cdots & ^{post}_{uni}F^1_{f5} \\ \vdots & \ddots & \vdots & \vdots & \ddots & \vdots & \vdots & \ddots & \vdots & \vdots & \ddots & \vdots \\ ^{pre}_{multi}F^m_{f1} & \cdots & ^{pre}_{multi}F^m_{f5} & ^{post}_{multi}F^m_{f1} & \cdots & ^{post}_{multi}F^m_{f5} & ^{pre}_{uni}F^m_{f1} & \cdots & ^{pre}_{uni}F^m_{f5} & ^{post}_{uni}F^m_{f1} & \cdots & ^{post}_{uni}F^m_{f5} \end{pmatrix}$$

$$m = 1, 2, \ldots 233$$

$$f1 \ldots f5 : 4, 8, 12, 16, 20\ kHz$$

To examine if frequency tuning curves were stable or plastic, similarity of the frequency tuning curves between pre- and post-learning sessions for each cell was quantified by Pearson's correlation (R) in auditory and multi-sensory trials separately. The distribution of Pearson's R pooled across all cells was compared between auditory and multi-sensory trials by ks-test (Supplementary Fig. 17c) to examine in which trial the frequency tuning curve would be more plastic or stable.

## Multi-sensory index

The multi-sensory index was calculated by the division of the average response in multi-sensory trials (AV) to the sum of average response in uni-sensory trials (A: auditory trials, V: visual trials). Cells with a non-significant response in both AV and A trials were excluded from multi-

trained on 70 % of the data and tested on a 30 % hold-out test-set. Same numbers of trials from each modality were selected for training and testing in 50 iterations to avoid a biased accuracy measurement due to unbalanced trial number in each modality (50 % multi-sensory, 16.7 % auditory, 13.3 % visual). For auditory frequency decoding (Supplementary Fig. 19c), trial sub-sampling was not necessary since each auditory frequency was presented for a same number of trials. Uni-sensory and multi-sensory trials were combined in one dataset. Visual stimulus feature was collapsed in multi-sensory trials leaving only frequency labels (4, 8, 12, 16 and 20 kHz). Train/test ratio remained 70/30 (140 training trials, 60 test trials). Simple accuracy (modality: x / 27 trials, frequency: x / 60 trials, x = number of correct decoded trials) in all iterations were averaged yielding pair-wise session decoding accuracy. Shuffle data ($n = 100$) were generated by circular permutation of the down-sampled time series features for each train/test iteration ($n = 50$) from which mean accuracy values of shuffle iterations were taken to depict chance decoding accuracy and further averaged to yield pair-wise session decoding accuracy (Supplementary Figs. 19b, c).

## Histology

After training, mice were transcardially perfused with phosphate buffer saline (PBS) followed by ca. 40 ml 4 % paraformaldehyde (PFA) in PBS. Immediately after perfusion, brains were removed and post-fixed in 4 % PFA overnight at 4 °C. Then, brains were stored in PBS at 4 °C until dissection. 150 μm coronal slices were prepared using a vibratome (Campden Instruments) and immunostained for calretinin as an anatomical marker as described previously[5]. Briefly, after PBS washes, brain slices were immersed in blocking solution (10% normal horse serum, S-2000-20, Vector Laboratories) with 0.5 % Triton (T8787, Sigma-Aldrich) in PBS for 2 h at room temperature. Next, slices were incubated in primary antibody (goat anti-calretinin, 1:1000, CG1, Swant) in carrier solution (1% normal horse serum with 0.5% Triton PBS) overnight at 4 °C. Slices were washed again in 0.5% Triton PBS and incubated for 2 h at room temperature or overnight at 4 °C in secondary antibody (donkey anti-goat 647, 1:1000, A-21447, Thermo-Fisher) in carrier solution. After final washes by PBS, slices were mounted on slides and cover slipped using 22 × 50 mm, 0.16–0.19 mm thick cover glass (FisherScientific). Images were acquired with a LSM700 confocal microscope (Zeiss), Axio Scan Slide Scanner (Zeiss) or Olympus BX63. Acquired images were post-processed with ImageJ (https://imagej.nih.gov/ij/) to locate the implantation site of the GRIN lenses.

## Statistical methods

Statistical analysis was performed in MATLAB (MathWorks). Alpha level was set at 0.05 and Bonferroni correction was applied to statistical tests (see Supplementary Table 1). Chi-square test, rank sum test, signed-rank test, linear mixed model (LMM), two-sample Kolmogorov-Smirnov test (ks-test2) and two-dimensional two-sample Kolmogorov-Smirnov test (https://github.com/brian-lau/multdist) were performed for datasets indicated in Supplementary Table 1. Data are presented as mean ± SEM unless otherwise stated. Statistical results and p values are presented in Supplementary Table 1.

## Reporting summary

Further information on research design is available in the Nature Portfolio Reporting Summary linked to this article.

## Data availability

Data to generate Figs. 1–4 and Supplementary Figs. 1–19 of this paper can be accessed in Source Data Main and Supplement, respectively. Statistical results are shown in Supplementary Table 1. The data generated in this study has been deposited in the German Neuroinformatics Node (https://doi.org/10.12751/g-node.7xxnmw). Source data are provided in this paper.

## Code availability

Custom-written code to generate figures and Supplementary Figs. has been deposited in the German Neuroinformatics Node (https://doi.org/10.12751/g-node.1rfzbn).

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

## Acknowledgements

We thank the HHMI Janelia GENIE Project for making jGCaMP7 available; Edward Boyden for making pAAV-hsyn-Jaws-KGC-GFP-ER2 available; Bryon Roth for making pAAV-hSyn-EGFP available; Raymond Strittmatter, Patrick Schlenker, Robert Häring and Simon Saner for mechanical workshop and electrical engineering support; all members of Gründemann laboratory for their general support and feedback on the manuscript. The research was supported by the following funding agencies and institutions: Swiss National Science Foundation (SNSF professorship PP00P3_170672, J.G.); European Research Council Starting Grant 803870 (J.G.); The Forschungsfonds Nachwuchsforschende of the University of Basel (M.H.); Forschungspreis der Schweizerischen Hirnliga (J.G.); The Department of Biomedicine at the University of Basel (J.G.); Deutsche Forschungsgemeinschaft (DFG), SFB 1089, SPP2411 (Teilprojekt, J.G.), Walter Benjamin Program Project 528405672 (R.P.M.); Deutsches Zentrum für Neurodegenerative Erkrankungen (DZNE), Bonn, Germany (J.G.); DZNE Innovative Minds Program (M.H.).

## Author contributions

Conceptualization: M.H. and J.G. Methodology: M.H., Z.H., R.P.M., and J.G. Investigation: M.H., Z.H., R.P.M., and J.G. Visualization: Z.H., M.H., R.P.M., and J.G. Funding acquisition: J.G., M.H., and R.P.M. Project administration: J.G. and M.H. Supervision: J.G. Writing – original draft, review & editing: M.H., Z.H., R.P.M., and J.G.

## Funding

## Competing interests

The authors declare no competing interests.
