## [Peer Review File · Nature Communications]

REVIEWER COMMENTS

Reviewer #1 (Remarks to the Author):

Hasegawa and colleagues use two-photon calcium imaging to measure MGB neuronal calcium dynamics within an associative learning and reversal learning task in head-fixed mice. They show learning-related encoding and adaptation in neuronal response dynamics that may or may not functionally contribute to learning. Overall, the data are well presented, the analyses are solid (with a few exceptions, see below), and the conclusions are very exciting. The most significant concern I have is that the meaning of the neuronal responses during learning have not been causally interpreted (see Major Concern #1). I suggest the authors add experiments to examine the causal contribution of these neurons for associative learning using temporally precise manipulations (i.e., optogenetics). This is a straightforward experiment that could be done independently of imaging and would significantly add to the interpretability of the manuscript.

Major Concerns

1. **Function:** An inherent assumption of the paper is that since MGB neurons encode learning and adapt across learning, these neurons likely contribute to learning. It would be very interesting to know if associative learning driven by auditory cues versus light cues is mediated by MGB. I suggest using optogenetics to inhibit MGB neurons during cue exposure.
2. **Single cell tracking:** First, it is not inherently obvious where the authors are using single cell tracking, versus showing untracked cells. Second, the authors do not provide examples of single cell tracking data, and thus it is difficult to visualize the accuracy and quality of the tracking. Third, the methods used for single-cell tracking are poorly described.
3. **Figure 1:** The claim that 'learning induced reward association of the Go stimulus during initial or reversal learning altered the proportion of stimulus responsive neurons toward the Go stimulus' is not substantiated by the simple chi-squared analysis performed in the figure caption, which only indicates that the proportion of all groups was different (not the direction, and not which groups are different) between naïve vs expert trials. The analysis needs to be improved.
4. **How do authors compare these data to other thalamic 2p calcium imaging datasets during associative or operant reward conditioning? Are MGB neurons less of a 'sensory relay' than previously expected?** This is the crux of the paper, and thus the interpretations need to be more broadly integrated with other studies involving thalamic sensory and non-sensory nuclei. For example, 2p imaging studies from PVT reveal neuronal response adaptations across learning that causally influence Pavlovian and operant reward conditioning (PMIDs: 31196673, 36369508). How might MGB neurons be working in a similar or unique fashion as compared with such thalamic substructures?

Moderate to Minor Concerns

1. Single cell clustering analysis could be more thoroughly shown. For example, how well does each cell 'fit' within each cluster? Can authors show silhouette plots indicating the fit of each cell for the given ensemble?
2. If 'MGB' abbreviation is defined in abstract, the actual name should be included rather than simply 'auditory thalamus'.
3. FA, Catch FA, and Hit are presented in the main figure but are not explained in the results or caption. This needs to be clearer so that readers do not have to scroll to the bottom of the paper to understand each measurement as presented. The same is true for d-Prime. Authors should be careful to be explicit about what each measurement/variable represents in in the results/figure captions.
4. Authors state in Line 66: "These data indicate that mice flexibly associate sensory stimuli with reward outcome across initial and reversal learning using similar learning strategies." The data don't necessarily suggest similar strategies are being used. In fact, we know different neuronal mechanisms are required for associative and subsequent reversal learning. Thus, this statement does not seem warranted. Rather, the data simply show that learning rates are similar.
5. Why do authors refer to MGB neurons as 'relay neurons'? Are they certain that all are 'relay neurons', and what exactly do they mean by that?

Reviewer #2 (Remarks to the Author):

In this study, the authors trained mice to perform Go/Nogo reversal learning task using counterbalanced auditory and visual stimuli as Go/NoGo cues, and tracked MGB neuronal calcium dynamics across learning using GRIN lens. Notably, they identified subsets of MGB neurons that are responsive to Go cues irrespective of sensory modalities. Moreover, a subset of such neurons demonstrated ramping activity during the delay period following sensory cues and before an action can be made (Fig. 2C and S3C). They did further analysis and concluded that MGB neurons developed more coherent population representation after learning, specifically in that delay period of Go trials. They also analyzed co-activity structure in MGB and showed that it is re-organized through learning and rule switches. Overall, their results suggest that the MGB may flexibly encode task-specific information, independently of auditory responsiveness, and expand our current understanding of this brain structure. The authors also conducted fair control experiments. However, there are several concerns that remain to be addressed.

1. In Fig.1 experiments, a more informative data to show would be the longitudinal changes in MGB responsiveness between initial and reversal learning phases. For example, the percentage of MGB neurons that are "Go cells" in both initial and reversal learning.
2. Identification of ramping activity in a subset of MGB neurons during the delay period of Go trials in Fig 2&S3 experiments is an intriguing finding. Is this activity correlated well with action outcomes and observed only in HIT but not MISS and FA trials?

3. Throughout their analysis, the author often pooled data from “auditory first” and “visual first” groups for statistics. However, given the known functions of the MGB, these two paradigms might not be perceived identically, and by assuming their equivalency, some crucial information might be obscured. I would expect more detailed comparative analyses for each paradigm to ensure more accurate and informative interpretation of the data.

In Fig. 1k&S2, the author did a proper job. From Fig. 1k that the proportion of “Go” cells in experts seem to predominate in both the initial and reversing learning phases. However, from Fig. S2, we can tell proportions of “NoGo” cells are actually predominant in experts of the reversal learning of “auditory first” group, and in experts of the initial training of “visual first group”, which makes sense given the sensory modalities and task contingency. By comparative analyses shown, the following conclusion is convincing:

76 Nevertheless, the learning-induced reward association of the Go stimulus during initial
77 or reversal learning altered the proportion of stimulus-responsive MGB neurons towards the
78 Go stimulus regardless of the sensory modality (Fig. 1k, Supplementary Fig. 2).

To reach the same quality, for Fig 2 experiments, I would suggest an equal number of mice for both groups that allow comparative analyses. In the right panel of Fig 2C, please color-code the data in the same format as Fig. 2A&B.

In Fig. S5B&C, for “Stim-Go” in the initial learning, PVC seems to increase for the expert in the visual group, though after pooling both groups there is no significant difference. In Fig. S6&S9, “Delay-go” in the reversal learning, the statistical difference seems to be contributed primarily from the auditory group. To further clarify, comparative analyses are warranted.

4. For PVC analyses (Fig. 3 and S5-9), the authors selected multiple sessions for each mouse. What are the selection criteria?

For Fig.3, while the authors’ effort in making a succinct layout is appreciated, it is very confusing for readers at first glance combining two independent matrices (triangles) into a square. Some additional diagram for clear illustration is highly suggested, or the author might just show the conventional ways as in Fig. S5-9.

5. 272 Co-activity network structure during the reward-preceding delay period was remodeled across
273 associative learning regardless of the sensory modality, while it remained by-and-large stable
274 during the stimulus period (Fig. 4b, Supplementary Figs. 10 and 12).

What is supporting evidence for the statement “regardless of the sensory modality”?

6. A minor issue: Fig. S2b lacks legends.

Reviewer #3 (Remarks to the Author):

Hasegawa et al., used deep-brain two-photon calcium imaging to record large populations of auditory thalamus (MGB) neurons while mice were performing a reversal learning task. They observed that MGB neurons are biased towards reward predictors independent of sensory modality. They found the MGB population response become more coherent across trials during the pre-reward delay period irrespective of the sensory modality. They conclude that network state changes in sensory thalamus represent learned outcomes.

The deep-brain imaging technique makes simultaneously recording a large population of MGB neurons possible. This study provided a new understanding towards the function of sensory thalamus. The conclusion of this study needs more experimental evidence and further analysis. My key concerns are the authors should rule out 1) the observed plasticity in MGB neurons is due to tissue damage caused by deep-brain imaging technique or change of GCaMP expression level over time, 2) the increased coherent activity is due to the general adaptation to the overall task environment.

Q1: Could the same neuronal populations be imaged and identified across all the experimental sessions reliably? Is there any cell loss in the imaging plane during the experimental period? Selection of cells included in the analysis could be a source of bias.

Q2: Since the conclusive results mainly focused on the comparison between naïve and expert phases, time is a nontrivial factor to consider. A big concern is, task-irrelevant factors such as GCaMP expression level, familiarity to the experimental environment, mice physiological states etc. anything co-varied with time along the recording sessions, could contribute a lot to the observed changing effects. It is not to say that the activity change or flexibility is solely caused by these irrelevant factors. However, confounded by them is not precluded. For example, familiarity with the task environment over time could alter certain aspects of mice behaviors (even if they were subtle) and the neuronal activity. In the case of PVC calculation, this time-progressive effect cannot be fully ruled out by taking No-Go and FA trials as controls, since they evoked quite different subgroups of neurons comparing to the Hit trials. Calculation of an overall PVC bleaches out the detailed info of which neurons contribute to the assumed learning-induced coherence.

Q3: If the network state changes in MGB represent learned outcomes, how coherent is the neuronal representation when the learning outcome is near the learning threshold? Could an increase of coherence of MGB population response be expected along with the increased learning performance?

Q4: Is the change of network state causally related to the functioning of associative learning? Can those network properties e.g. hubness, clustering coefficients, predict how well the animal learned?

Q6: It seems, no matter from the perspective of behavioral results or the neuronal activity, the mice

performed the reversal learning as a completely new task just as it did in the initial learning, for example, no learning effect such as an increased learning rate was shown/found to transfer from the initial learning to the reversal learning. If that was the case, then, it's very likely that the mice learned two tasks independently. Considering that, either the statements should focus on reward-associative learning (rather than reversal learning), or further analysis should be conducted to reveal the difference between the initial and the reversal learning. What makes "reversal" different from a sole learning? What has been rebuild exactly?

We would like to thank all reviewers for their helpful and constructive comments on the manuscript. We have now included the reviewers' suggestions for additional analysis and experiments, including inhibitory optogenetics to demonstrate that MGB neurons are causally linked to the learning of the audiovisual reward-associative task.

REVIEWER COMMENTS

Reviewer #1 (Remarks to the Author):

Hasegawa and colleagues use two-photon calcium imaging to measure MGB neuronal calcium dynamics within an associative learning and reversal learning task in head-fixed mice. They show learning-related encoding and adaption in neuronal response dynamics that may or may not functionally contribute to learning. Overall, the data are well presented, the analyses are solid (with a few exceptions, see below), and the conclusions are very exciting. The most significant concern I have is that the meaning of the neuronal responses during learning have not been causally interpreted (see Major Concern #1). I suggest the authors add experiments to examine the causal contribution of these neurons for associative learning using temporally precise manipulations (i.e., optogenetics). This is a straightforward experiment that could be done independently of imaging and would significantly add to the interpretability of the manuscript.

Major Concerns

1. Function: An inherent assumption of the paper is that since MGB neurons encode learning and adapt across learning, these neurons likely contribute to learning. It would be very interesting to know if associative learning driven by auditory cues versus light cues is mediated by MGB. I suggest using optogenetics to inhibit MGB neurons during cue exposure.

The causal role of activity in MGB was not explored in the original manuscript. We have now added experiments with optogenetic inhibition of MGB activity during the sensory stimulus presentation. We bilaterally expressed either the inhibitory opsin, Jaws (Chuong et al., 2014), (AAV2/5-hsyn-Jaws-KGC-GFP-ER2 from Addgene, 65014-AAV5) or GFP (AAV2/5-hSyn-EGFP from Addgene, 50465-AAV5) in MGB. During Go (auditory) / Nogo (visual) training, the cue periods were paired with LED stimulation (617 nm) of MGB via bilateral optical fibers. To avoid rebound activity at the end of optogenetic inhibition, the intensity of the LED light was ramped down over 250 ms. In contrast to the control group, the inhibitory opsin-expressing group did not reach the d' threshold (min 1.5) after 10 days of training, indicating significantly slower learning than the control group (Rebuttal Fig. 1). These results demonstrate that inhibition of MGB impairs learning suggesting that the activity of auditory thalamus neurons is causally linked to associative reward learning in our paradigm.

Rebuttal Fig. 1 | Optogenetic inhibition of MGB activity perturbs associative stimulus-reward learning. (a) Left: d-prime across learning (mean +/- SEM). Right: d-prime of individual mice. Each line represents the d-prime values of one mouse. d-primes of the opsin-positive group were significantly lower than those of the GFP groups at the end of the training (Session 9 & 10, $p = 0.0230$ and $p = 0.0067$ respectively, Two-way ANOVA with Sidak's multiple comparison test). (b) d-prime of individual mice in Session 10.

These new data are now added to the main text (Fig.1h) and supplementary material (Supplementary Fig. 2). The new experimental procedures are described in the methods section (“Surgical procedures” and “Optogenetic inhibition”).

2. *Single cell tracking: First, it is not inherently obvious where the authors are using single cell tracking, versus showing untracked cells. Second, the authors do not provide examples of single cell tracking data, and thus it is difficult to visualize the accuracy and quality of the tracking. Third, the methods used for single-cell tracking are poorly described.*

We have now made it more explicit in the text to highlight if tracked vs. untracked cells were used in the analysis. In summary, tracked cell data was used for k-mean clustering (Fig. 2. Supplementary Fig. 4), co-activity network analysis (Fig. 4, Supplementary Fig. 12 – 14), correlation analysis between neural data and behavioral variables (Supplementary Fig. 6) and the sensory mapping analysis (Supplementary Fig. 15-19). Untracked cell data was used for all other analyses. This is now also highlighted in each respective paragraph of the main text and the Methods section.

Furthermore, we now added examples of representative tracked single cells (single cell images and calcium traces, Rebuttal Fig. 2 and Supplementary Fig. 4a).

Rebuttal Fig. 2 | Examples of individual tracked cells. Image planes of four example cells and their respective calcium traces across learning from one mouse. MGB neurons were successfully tracked across learning for several weeks and their activity changed dynamically across reward-associative learning. Black and gray lines on each calcium trace represent stimulus presentation (2 s) and delay period (2 s), respectively.

We now added a more extensive description (see Methods) of the cross-session single cell registration procedure. Briefly, we used two-photon images from the first imaging session as a reference template. In the following sessions, we returned to this template image plane at the beginning of the two-photon imaging session. To return to the same imaging plane, we used several cells with bright signals and clear contours as reference cells. Angular rotation was prevented through a custom design of a tight-fitting head bar.

To extract calcium signals from the tracked cells across imaging sessions, we used “RoiMatchPub” (<https://github.com/ransona/ROIMatchPub>), as described in the Methods section (“Longitudinal cell tracking and signal extraction”). RoiMatchPub is a program to match the data of regions of interest (ROI) across sessions. This program handles the ROI data generated by Suite2P (<https://github.com/MouseLand/suite2p>, see also, <https://suite2p.readthedocs.io/en/latest/multiday.html>). RoiMatchPub uses a semi-automatic cell tracking algorithm. At first, we prepared a template two-photon image, e.g., the max-intensity projection (MIP) two-photon image acquired in the first imaging session, and the MIP image of the following sessions. Then, we manually matched as many of the same ROIs as possible in the General User Interface of RoiMatchPub between the first template two-photon image and two-photon images in

the following sessions (e.g., 1st session image – 2nd session image, 1st session image – 3rd session image, 1st session image and 4th session image and so on).

3. Figure 1: The claim that ‘learning induced reward association of the Go stimulus during initial or reversal learning altered the proportion of stimulus responsive neurons toward the Go stimulus’ is not substantiated by the simple chi-squared analysis performed in the figure caption, which only indicates that the proportion of all groups was different (not the direction, and not which groups are different) between naïve vs expert trials. The analysis needs to be improved.

We have now added an alternative analysis approach based on the proportion of Go-responsive neurons for each mouse before and after learning (Rebuttal Fig. 3). The proportion of Go-responsive cells increases from naïve to expert stages in both initial and reversal learning on a mouse-by-mouse level ($p < 0.01$, signed-rank-test). This result confirms our population-level observation that stimulus-reward associative learning altered the proportion of stimulus responsive neurons toward the Go stimulus. We have now incorporated this new analysis as Fig. 1m. We would like to mention that the proportion of sensory responsive cells across learning in each mouse is also shown in Supplementary Fig. 3b.

Rebuttal Fig. 3 | The proportion of Go-responsive cells significantly increases across learning. Left: Proportion of Go stimulus-responsive cells of each mouse in initial learning. Right: Proportion of Go stimulus-responsive cells of each mouse in the reversal learning. The proportions of Go stimulus-responsive cells were significantly increased from the naïve to the expert phase across learning in both initial and reversal learning ($p < 0.05$ for initial and reversal learning, signed-rank test). This analysis is now added as new Fig. 1m.

4. How do authors compare these data to other thalamic 2p calcium imaging datasets during associative or operant reward conditioning? Are MGB neurons less of a ‘sensory relay’ than previously expected? This is the crux of the paper, and thus the

interpretations need to be more broadly integrated with other studies involving thalamic sensory and non-sensory nuclei. For example, 2p imaging studies from PVT reveal neuronal response adaptations across learning that causally influence Pavlovian and operant reward conditioning (PMIDs: 31196673, 36369508). How might MGB neurons be working in a similar or unique fashion as compared with such thalamic substructures?

The reviewer is right. This is an important point to add. We have now elaborated on additional substructures in thalamus and their potential (combined) role in associative learning and outcome prediction.

In general, we don't think that auditory thalamus is "less of a 'sensory relay'" (e.g., we find stable sensory representations on the level of subtypes of individual neurons as well as the population level, Fig. 2 and Supplementary Fig. 19). Nevertheless, thalamus plays a fundamental role in addition to its sensory 'relay' functions including modulation of sensory information, learning- and behavior-related plasticity as well as representation of outcomes and task-related features (as shown in this study). Specifically, we think that MGB encodes and broadcasts learning- and behaviorally-relevant information to other brain regions across sensory reward-associative learning that goes beyond a classical sensory information relay function and thus contributes to a wider network of memory formation and processing of salient information. This suggests that MGB balances sensory relay and cognitive functions.

We added this notion to the discussion:

"The complementary mechanisms of single cell plasticity and population-level stability of sensory coding could be crucial to allow for dynamic neuronal representations upon learning, while ensuring stable representations of the environment (Betz et al., 2019; Taylor et al., 2021), balancing relay and cognitive functions of MGB."

In addition to sensory thalamic nuclei, thalamic nuclei outside of the classical sensory pathways exhibit adaptive responses to reward-associated sensory stimuli. As the reviewer points out, neurons in the paraventricular nucleus of the thalamus (PVT) develop adaptive responses upon cue-reward learning in a circuit specific fashion (PMIDs: 31196673, 36369508). In addition, work on the posteromedial nucleus of the thalamus (POm) also reports that POm encodes behavioral outcomes; particularly correct response information in the expert phases of goal-directed tactile behavior (La Terra et al., 2022). Finally, a recent preprint (Petty & Bruno, 2024) demonstrates that POm activity increases to sensory stimuli associated with reward, indicating that POm encodes outcome predictors. These changes in POm are regardless of sensory modality (tactile or visual stimulus), as we observed in MGB. Currently, there is no conclusive model how different first order, higher order and 'non-sensory' thalamic nuclei interact to shape learning and memory and future studies probing information flow and causality will be necessary to delineate the role of this distributed thalamic code.

We have added these points to the discussion as follows:

"Increasing evidence driven by dense electrophysiological recording techniques and deep brain imaging of identified cell types in head-fixed or freely moving animals reveals diverse functions of individual neurons in different thalamic nuclei"

during learning and adaptive behavior. For example, neurons in paraventricular thalamus exhibit heterogeneous response adaptation during cue-reward learning and reward seeking, which are suppressed by fearful stimuli (Otis et al., 2019; Vollmer et al., 2022). Similarly, neuronal responses in MGB are adaptive to reward- (see above) and aversive-outcome predicting sensory stimuli during associative learning (Barys et al. 2020, Taylor et al., 2021). Furthermore, sensory responses in higher order somatosensory and visual thalamus are modulated by attention and reward-predicting stimuli even by non-classical modalities similar to our findings (La Terra et al., 2022; Petty & Bruno, 2024). Despite these response similarities, it is currently unclear if and how distinct thalamic areas including first order ‘relay nuclei’ as well as higher-order and non-sensory thalamus interact together with cortex and other brain areas to shape learning. Multisite large-scale recording approaches (Chen et al., 2024; Toader et al., 2023) and (all-optical) perturbations of information flow between thalamic subnuclei and their non-thalamic inputs and outputs might help to reveal the distributed population code during adaptive behavior.”

Moderate to Minor Concerns

1. Single cell clustering analysis could be more thoroughly shown. For example, how well does each cell ‘fit’ within each cluster? Can authors show silhouette plots indicating the fit of each cell for the given ensemble?

As suggested, we generated silhouette plots and silhouette scores to assess the goodness of sorting for each cell within the respective clusters (Rebuttal Fig. 4). The mean silhouette score across clusters is 0.331, similar to recent work (Grant et al., 2021) comparing clustering models to sort medial prefrontal cortex neurons into functional clusters. Although we used a k-means algorithm, our silhouette score (0.331) is comparable to the score of an alternative method (spectral clustering, 0.38) (Grant et al., 2021). We must note that our clustering method is semi-manual, i.e. similar functional clusters with slight differences in intra-event temporal dynamics, yet similar changes across learning, got merged by hand and thus can dilute the silhouette scores.

Rebuttal Fig. 4 | Silhouette scores of MGB neurons of each functional cluster. MGB neurons were sorted into functional clusters by a k-means clustering algorithm and post hoc manually merged for

functionally similar clusters. Silhouette scores represent the goodness of sorting of each MGB neuron into the clusters.

2. If 'MGB' abbreviation is defined in abstract, the actual name should be included rather than simply 'auditory thalamus'.

We have added the full name now in the abstract.

3. FA, Catch FA, and Hit are presented in the main figure but are not explained in the results or caption. This needs to be clearer so that readers do not have to scroll to the bottom of the paper to understand each measurement as presented. The same is true for d-Prime. Authors should be careful to be explicit about what each measurement/variable represents in in the results/figure captions.

Thank you very much for your suggestion. This will improve the legibility and we added the explanation of behavioral responses, their outcomes and d-prime in the legend of Figs. 1c and 1d.

4. Authors state in Line 66: "These data indicate that mice flexibly associate sensory stimuli with reward outcome across initial and reversal learning using similar learning strategies." The data don't necessarily suggest similar strategies are being used. In fact, we **know different neuronal mechanisms are required for associative and subsequent reversal learning**. Thus, this statement does not seem warranted. Rather, the data simply show that **learning rates** are similar.

We agree with the reviewer. The primary focus of the paper is not reversal learning and there might be alternative explanations why learning rates are similar. We have removed the clause about similar learning strategies to avoid misunderstandings.

5. Why do authors refer to MGB neurons as 'relay neurons'? Are they certain that all are 'relay neurons', and what exactly do they mean by that?

To avoid ambiguity, we removed the term 'relay'.

Reviewer #2 (Remarks to the Author):

In this study, the authors trained mice to perform Go/Nogo reversal learning task using counterbalanced auditory and visual stimuli as Go/NoGo cues, and tracked MGB neuronal calcium dynamics across learning using GRIN lens. Notably, they identified subsets of MGB neurons that are responsive to Go cues irrespective of sensory modalities. Moreover, a subset of such neurons demonstrated ramping activity during the delay period following sensory cues and before an action can be made (Fig. 2C and S3C). They did further analysis and concluded that MGB neurons developed more coherent population representation after learning, specifically in that delay period of Go trials. They also analyzed co-activity structure in MGB and showed that it is re-organized through learning and rule switches. Overall, their results suggest that the MGB may flexibly encode task-specific information, independently of auditory responsiveness, and expand our current understanding of this brain structure. The

authors also conducted fair control experiments. However, there are several concerns that remain to be addressed.

1. In Fig.1 experiments, a more informative data to show would be the longitudinal changes in MGB responsiveness between initial and reversal learning phases. For example, the percentage of MGB neurons that are “Go cells” in both initial and reversal learning.

Fig. 1 reports the number of cells that are Go responsive after each learning period. Given that the sensory stimuli are of different modality, it is striking that the number of Go responsive cells in auditory thalamus increases irrespective of the modality. Fig. 1 includes all cells that could be recorded in each session and demonstrates that the responsiveness of individual MGB neurons, on average, is biased towards the Go stimuli after learning. Fig. 1 is therefore a general description of the population activity in MGB.

Fig. 2 analyzes all cells that could be tracked across different stages of the paradigm and allows for a comparative analysis of adaptive changes in response patterns of distinct plasticity types in each learning phase. We believe that this illustrates the point that the reviewer is asking for: Using tracked cell data (n = 210 neurons), we found that the proportion of stable Go-responsive cells, i.e. Cells which always respond to the Go stimulus irrespective of the modality and without changes in activity patterns across all phases (Initial Naïve, Initial Expert, Reversal Naïve, Reversal Expert), is ca 1%. This is not unexpected as these cells need to be multisensory and stable across learning irrespective of the learning phase. In addition, and this is already shown in the paper, and we believe that this is what the reviewer asks for, a fraction of MGB neurons (Stable-up cluster, 5 %) is consistently potentiated to the Go-stimulus in the expert phases in initial and reversal learning, irrespective of the modality (Fig. 2c). Thus, a relatively small fraction of MGB neurons is consistently responsive to cross-modal Go-stimuli regardless of the learning phase, expert level and sensory modality, yet a substantial fraction of MGB neurons is adaptive to cross-modal Go stimuli, both modality-independent and driven by learning.

2. Identification of ramping activity in a subset of MGB neurons during the delay period of Go trials in Fig 2&S3 experiments is an intriguing finding. Is this activity correlated well with action outcomes and observed only in HIT but not MISS and FA trials?

This is an interesting point, and we now provide a new figure further analyzing the ramping cell activity (Ramp-up and Ramp-down cells) in Hit and FA trials. As shown below (Rebuttal Fig. 5) and as a new supplementary figure (Supplementary Fig. 5), the activity of ramping cells is correlated with task outcome, but not the action, since their unique activity patterns are observed only in Hit trials. Interestingly, in FA trials, ramp-up cells exhibit increased activity in the naive phase of the reversal period. This may be due to the paradigm structure, i.e. mice predicted the reward in the Nogo trials after reversal, which was the previous Go trial in initial learning but remain now unrewarded and the animal has to renew the cue-reward contingency, which leads to abolished ramping-responses to previous now-unrewarded Go cues. This result also supports our finding that ramping cells represent predicted outcomes.

Please note that this analysis can only be conducted for Hit and FA trials, but not Miss trials, since the number of Miss trials is too small or none-existing in the expert stages.

Rebuttal Fig. 5 | Activity of ramp-up and ramp-down cells in Hit and FA trials. Both, ramp-up and ramp-down cells show unique ramping activity patterns only in Hit trials, while their activity patterns in FA trials are distinct from those in Hit trials. Ramp-up cells show increased activities in FA trials in the naive phase in reversal learning. Black and gray lines in each graph represent stimulus (2 s) and delay period (2 s), respectively.

3. Throughout their analysis, the author often pooled data from “auditory first” and “visual first” groups for statistics. However, given the known functions of the MGB, these two paradigms might not be perceived identically, and by assuming their equivalency, some crucial information might be obscured. I would expect more detailed comparative analyses for each paradigm to ensure more accurate and informative interpretation of the data.

In Fig. 1k&S2, the author did a proper job. From Fig. 1k that the proportion of “Go” cells in experts seem to predominate in both the initial and reversing learning phases. However, from Fig. S2, we can tell proportions of “NoGo” cells are actually predominant in experts of the reversal learning of “auditory first” group, and in experts of the initial training of “visual first group”, which makes sense given the sensory modalities and task contingency. By comparative analyses shown, the following conclusion is convincing: Nevertheless, the learning-induced reward association of the Go stimulus during initial or reversal learning altered the proportion of stimulus-responsive MGB neurons towards the Go stimulus regardless of the sensory modality (Fig. 1k, Supplementary Fig. 2).

I would suggest an equal number of mice for both groups that allow comparative analyses.

In the right panel of Fig 2C, please color-code the data in the same format as Fig. 2A&B.

We would like to thank the reviewer for the positive comments and suggestion.

Overall, we included 8 mice in the study where we could image cells consistently across the long reversal paradigm. We carefully chose to counterbalance the order of the associated stimuli during the learning paradigm. We employed conservative selection criteria to include cells, particularly for the across-day alignment of cells (see also comments to reviewer 1). This resulted in a reduction of animal numbers to N = 6 mice (4 animals auditory first, 2 animals visual first) for main Figures 2 and 4. Although this resulted in a slight imbalance of the orders of modalities, we preferred this solution to inducing potential ambiguities in aligned cells. Nevertheless, we find that the overall conclusions for Figs. 2 and 4 hold irrespective of the imbalance in modality. We added further analysis to test for potential effects of the modality on the results of the PVC analysis. Here we got an additional result. Interestingly, for the stimulus period of the visual group upon initial learning, we found that the correlation significantly increased from naive to expert (Supplementary Fig. 5b, now Supplementary Fig. 7c, left panel, $p=0.019$, linear mixed model) in addition to the delay period. This result is interesting and might indicate that, at least during initial learning, the stimulus responses of MGB to visual stimuli become more coherent, which is not the case for auditory stimuli. This data suggests that the ‘non-preferred’ modality shows more coherent representations in MGB when paired with a reward, while the canonical auditory modality is not affected.

Regarding the right panel of Fig. 2c, these data are sampled from both, the initial (I) and Reversal (R) periods and can thus inherently include cells that are responsive to either stimulus, the auditory and the visual stimulus. To color code Fig. 2c similar to Fig. 2a,b, we would have to assign dual colors to the individual plots, or a more complex color scheme, which will make the figure harder to understand. However, in line with the reviewers general idea of point 3, we are now providing a new Rebuttal Fig. 6, which doesn't color code the proportion of cells by auditory or visual responsiveness, as Fig. 2a,b, but by modality order cohort, i.e. Auditory-first or Visual-first.

Rebuttal Fig. 6 | Proportion of cells in functional subgroups exhibiting task and modality-specific plasticity in the two expert phases in initial and reversal learning. This figure is modified from the

right panel of Fig. 2c. The order of stimulus-reward contingency is color-coded. "Auditory-first" group (red dots) was trained to associate the auditory stimulus with reward in the initial learning and trained to associate the visual stimulus with reward in the reversal learning. "Visual-first" group (green dots) was trained to associate sensory stimulus with reward in a counter-balanced manner.

Similar to the PVC results, we do not observe an obvious bias of the modality order of the paradigm on the proportion of distinct cell types. We hope that this additional rebuttal figure dispels the reviewer's concerns.

For practical reasons, and if the reviewer agrees, we would like to avoid adding this color code to main Fig. 2c, given that the color codes of Fig. 2a,b are used for a different purpose, i.e. indication of reward-association to individual modality and not modality order of the paradigm. We feel that this additional color code would add confusion to the overall flow of the figure as it addresses a separate issue.

In Fig. S5B&C, for "Stim-Go" in the initial learning, PVC seems to increase for the expert in the visual group, though after pooling both groups there is no significant difference. In Fig. S6&S9, "Delay-go" in the reversal learning, the statistical difference seems to be contributed primarily from the auditory group. To further clarify, comparative analyses are warranted.

We performed additional statistical tests for the Go 'Delay' and 'Stimulus' periods of the visual group.

For the Delay period upon reversal learning, we found that R values in the visual reward group significantly increased from naive to expert upon reversal learning (Supplementary Fig. 6, now Supplementary Fig. 8, $p=0.036$, linear mixed model). This confirms that the data from both auditory and visual groups contribute to the statistical difference of the PVC.

Supplementary Fig. 9 (now, Supplementary Fig. 11) focusses on the contribution of ramping cells to the PVC. We might misunderstand the reviewer's concern, but the analysis tests if specific subsets of cells (i.e. ramping-cells shown in Fig. 2) are solely contributing to the observed increase in PVC during the delay period upon learning. If ramping-cells drove the PVC changes during the delay periods, we would see a drop in the effect, irrespective of the initial modality. This is not the case.

Interestingly, for the stimulus period of the visual group upon initial learning, we found that the correlation significantly increased from naive to expert (Supplementary Fig. 5b, now, Supplementary Fig. 7c, left panel, $p=0.019$, linear mixed model). This result is interesting and might indicate that, at least during initial learning, the stimulus responses of MGB to visual stimuli become more coherent, which is not the case for auditory stimuli. This data suggest that the 'non-preferred' modality shows more coherent representations in MGB when paired with a reward, while the canonical auditory modality is not affected.

4. For PVC analyses (Fig. 3 and S5-9), the authors selected multiple sessions for each mouse. What are the selection criteria?

For Fig. 3 and Supplementary Figs. 5-9 (now Supplementary Figs. 7-11), we selected 2-3 sessions from naive and expert stages. The expert stage is defined based on the discriminability index, d-prime. The naive phase consisted of the first 2-3 behavioral sessions in which the d-prime was less than 1.5 (learning threshold). The expert phase consisted of at least three consecutive behavioral sessions in which d-prime passed the threshold of 1.5. For this reason, we selected three sessions in both naive and expert phases for our analysis. Some mice learned the task quickly and d-prime already reached over 1.5 in the third behavioral session in the naive stage. In this case, we have two sessions for the analysis in naive stages. One mouse reached only two reversal expert sessions before the training had to be terminated. In this case, we have only two sessions for our analysis in the expert phase (Methods, "Behavioral training in a sensory Go/Nogo reversal learning task"). This explanation is now added to the methods section.

For Fig.3, while the authors' effort in making a succinct layout is appreciated, it is very confusing for readers at first glance combing two independent matrices (triangles) into a square. Some additional diagram for clear illustration is highly suggested, or the author might just show the conventional ways as in Fig. S5-9.

We now added extra white space and new label colors in the Fig. 3a-d to avoid confusion of the asymmetric matrices by preserving the succinct layout in Fig. 3 yet allow for additional clarity. We hope that this new design will enhance the distinction between the two analysis periods.

5. 272 Co-activity network structure during the reward-preceding delay period was remodeled across

273 associative learning regardless of the sensory modality, while it remained by-
and-large stable

274 during the stimulus period (Fig. 4b, Supplementary Figs. 10 and 12).

What is supporting evidence for the statement "regardless of the sensory modality"?

To ensure that the network structure during the delay period changed in a similar manner in both auditory and visual reward conditions, we performed a separate analysis (similar to Fig. 4) based on whether the animals were first trained on auditory or visual Go cues, respectively. The trend of increased hubness and decreased path length is similar in animals that were trained first on auditory or visual Go cues. (Rebuttal Fig. 7).

Rebuttal Fig. 7 | Co-activity network in Go-trial delay period of auditory-Go (initial) and visual-Go (initial) group. Both groups share a similar trend of change of network structure as shown in Fig 4. Error bars show 95% confidence interval of mean.

6. A minor issue: Fig. S2b lacks legends.

Thank you! We have now added the legends for Fig. S2b (now new Supplementary Fig. 3b).

Reviewer #3 (Remarks to the Author):

Hasegawa et al., used deep-brain two-photon calcium imaging to record large populations of auditory thalamus (MGB) neurons while mice were performing a reversal learning task. They observed that MGB neurons are biased towards reward predictors independent of sensory modality. They found the MGB population response become more coherent across trials during the pre-reward delay period irrespective of the sensory modality. They conclude that network state changes in sensory thalamus represent learned outcomes.

The deep-brain imaging technique makes simultaneously recording a large population of MGB neurons possible. This study provided a new understanding towards the function of sensory thalamus. The conclusion of this study needs more experimental evidence and further analysis. My key concerns are the authors should rule out 1) the

observed plasticity in MGB neurons is due to tissue damage caused by deep-brain imaging technique or change of GCaMP expression level over time, 2) the increased coherent activity is due to the general adaptation to the overall task environment.

1) the observed plasticity in MGB neurons is due to tissue damage caused by deep-brain imaging technique or change of GCaMP expression

We would like to thank the reviewer for the positive comments on our study.

Regarding the reviewer's first concern, we do not think that tissue damage or changes of GCaMP expression might be the cause of functional plasticity of MGB neurons.

First, the responses are time locked to the behavioral task variables, demonstrating that Ca^{2+} changes reflect a functional response instead of fluorescence drift or cell damage.

Second, the observed reward-associative plasticity was learning-specific to the Go stimulus (Fig. 1l and m). This is similar to our previous work in the aversive domain demonstrating that MGB plasticity is learning-specific to aversive associated stimuli and not just a general drift over time (Taylor et al., 2021).

Third, the learning paradigm is reversible and the responsiveness of Go and Nogo activity patterns shift in a task-dependent manner. Our study demonstrates that MGB plasticity occurs from naive to expert stages in both initial and reversal learning in a learning-related fashion. If the response plasticity would solely reflect changes in GCaMP expression level in later learning stages, we would not observe (reversible) MGB plasticity in the expert stages of reversal learning. Similarly, if the cells would just load with Ca^{2+} over time or bleach we would observe unidirectional changes and not reversible, task-specific changes in activity patterns, that can be both, excitatory or inhibitory.

Together, we think that it is safe to exclude that the stimulus- and outcome-specific changes in MGB plasticity can be attributed to tissue damage or increased Ca^{2+} sensor expression, which would lead to non-specific changes and loss of responsiveness.

Ca^{2+} imaging is a standard technique for longitudinal activity measurements in cortex over long time periods and our data (together with others recording e.g. in PVT, hypothalamus or hippocampus) shows that similar imaging studies can be performed in deep brain structures via GRIN lenses without tissue damage impacting neuronal activity.

We would also like to refer to Rebuttal Fig. 2 (redrawn below), which shows examples of stable somatic Ca^{2+} expression and activity of representative neurons across the imaging and learning paradigm. The somatic Ca^{2+} expression is stable over weeks and the neuronal activity changes dynamically across learning. We do not observe saturated signals. This rebuttal figure is now part of Supplementary Fig. 4.

Rebuttal Fig. 2 | Examples of individual tracked cells. Image planes of four example cells and their respective calcium traces across learning from one mouse. MGB neurons were successfully tracked across learning for several weeks and their activity changed dynamically across reward-associative learning. Black and gray lines on each calcium trace represent stimulus presentation (2 s) and delay period (2 s), respectively.

2) *the increased coherent activity is due to the general adaptation to the overall task environment.*

We show that the MGB changes in network activity are predictive of the task outcome and reverse with a changing stimulus contingency in a reversible manner. Given this dynamic change in relation to the stimulus, we think that the coherent activity cannot be explained by adaptation to the task environment, which would result in linear changes over time but not reversible task-specific changes.

Q1: Could the same neuronal populations be imaged and identified across all the experimental sessions reliably? Is there any cell loss in the imaging plane during the experimental period? Selection of cells included in the analysis could be a source of bias.

The same imaging planes could be recorded across all analysis sessions. In those imaging planes, we identified 837 (naive, initial learning), 755 (expert, initial learning), 808 (naive, reversal learning) and 709 cells (expert, reversal learning) across learning (Fig. 1j, k). Among these cells, we successfully tracked 210 cells across the reversal learning paradigm. Thus, we sampled approximately 25%, 28%, 26% and 30% of the

total population as the tracked cell in each learning stage. This subsampling is a conservative approach to ensure that only reliably tracked cells were included in the analysis of Fig. 2 and 4. Inherently, there is always some cell loss in the imaging planes, and we made sure to apply conservative criteria to match the cells across days to favor confidence in cell matching over false-positive matched cells. This naturally introduces a bias towards cells that can be matched, although we do not believe that this introduces a biological bias in our results and rather strengthens the quality of the data.

Q2: Since the conclusive results mainly focused on the comparison between naïve and expert phases, time is a nontrivial factor to consider. A big concern is, task-irrelevant factors such as GCaMP expression level, familiarity to the experimental environment, mice physiological states etc. anything co-varied with time along the recording sessions, could contribute a lot to the observed changing effects. It is not to say that the activity change or flexibility is solely caused by these irrelevant factors. However, confounded by them is not precluded. For example, familiarity with the task environment over time could alter certain aspects of mice behaviors (even if they were subtle) and the neuronal activity. In the case of PVC calculation, this time-progressive effect cannot be fully ruled out by taking No-Go and FA trials as controls, since they evoked quite different subgroups of neurons comparing to the Hit trials.

Given that the learning paradigm and stimulus contingencies were reversed after the initial learning period and the PVC changes follow this reversal, we think that time, adaptation or familiarity to the environment are not driving factors of the overserved changes in PVC. The reversal of the stimulus-reward contingency accounts for that. Furthermore, the mice are counterbalanced for stimulus-reward association, and we observe the same effects irrespective of the nature of the go cue in initial learning (see also comments to Reviewer 2). Finally, the delay period (where we observe the outcome dependent effects in Hit trials) is void of any stimulation and the same across all trial types (2 seconds between end of cue and start of spout movement) such that changes in GCaMP expression or other time-progressive effects cannot account for the learning- and trial-specific effects that we observe in MGB.

Regarding a discussion of GCaMP expression levels please see our answer to *Point 1* above.

Calculation of an overall PVC bleaches out the detailed info of which neurons contribute to the assumed learning-induced coherence.

Yes, the PVC calculates a population level effect and does not consider individual neurons. Nevertheless, given the strong prominence of ramping neurons in the delay periods, we also calculated the PVC during the delay period after removing this specific subgroup of neurons to understand their effect on the total PVC (Supplementary Fig. 11). Removal of random subgroups of neurons served as a control. Nevertheless, even after removing ramping neurons from the PVC calculation, the PVC increase from naïve to expert animals is consistent, indicating that whole MGB population, and not individual subtypes of neurons, contribute to the population level change across learning (Supplementary Fig. 11).

Q3: If the network state changes in MGB represent learned outcomes, how coherent is the neuronal representation when the learning outcome is near the learning threshold? Could an increase of coherence of MGB population response be expected along with the increased learning performance?

To address the question, we analyzed the relationship between the behavioral task performance and PVC. Fitting a linear model for the data pooled from initial and reversal learning, we observed a positive linear correlation of performance and the strength of the PVC during the delay period (Rebuttal Fig. 8). This data indicates that an increase in coherence could potentially be expected along with gradual changes in task performance. Nevertheless, since the manuscript in its current form is not focused on neuronal coding at transition periods of behavioral performance at learning threshold, we think that an addition of this initial observation would not be in the scope of this study.

Rebuttal Fig. 8 | PVC is positively correlated with the task performance. A linear relationship was found between PVC during the delay period and performance (slope = 0.0022, $p < 0.01$). Each dot represents individual session data from naive and expert stages in initial and reversal training.

Q4: Is the change of network state causally related to the functioning of associative learning? Can those network properties e.g. hubness, clustering coefficients, predict how well the animal learned?

On the level of network states, we did not observe a correlation between the performance of the animals and the changes within the network (Rebuttal Fig. 9), despite the changes being predictive of the outcome and the positive correlation between the PVC and the task performance (Rebuttal Fig. 8). We think that this points towards a more immediate function of the changes in network correlations and their relation to trial-by-trial task outcome than the overall trial-by-trial learning performance level of the animal, which seems to be reflected in the overall PVC. However, at the moment it will be a hard experiment to perform, since it would require the induction of specific network states, which could not be tested with all-optical approaches and stimulation of functional subpopulations of neurons (given that the contribution of individual identified functional subgroups alone, i.e. ramping cells, to the observed network state changes is weak, Supplementary Fig. 12).

Rebuttal Fig. 9 | No obvious linear correlation was found between co-activity structure and performance. Co-activity network structure during the delay period does not predict the performance of an animal. Each dot is the mean value of path length, hubness and cluster coefficient in a session including initial and reversal training.

Q6: It seems, no matter from the perspective of behavioral results or the neuronal activity, the mice performed the reversal learning as a completely new task just as it did in the initial learning, for example, no learning effect such as an increased learning rate was shown/found to transfer from the initial learning to the reversal learning. If that was the case, then, it's very likely that the mice learned two tasks independently. Considering that, either the statements should focus on reward-associative learning (rather than reversal learning), or further analysis should be conducted to reveal the difference between the initial and the reversal learning. What makes "reversal" different from a sole learning? What has been rebuild exactly?

We agree with the reviewer. A similar point has been raised by reviewer 1. The aim of the study was not to focus on the neural mechanism of reversal learning per se in MGB but to understand neuronal coding of reward-associative learning in MGB and its cross-modal character. We have found that MGB neurons flexibly encode the stimulus and behavioral outcome across our behavioral paradigm in a cross-modal fashion.

On the behavioral level, we did not observe increased learning rates after reversal similar to classical reversal learning, as observed e.g. by Banerjee et al., 2020. This could be due to differences in task design. In our paradigm, the reward contingency was changed in a cross-sensory manner to test if MGB plasticity is generalizable or different for distinct modalities in reward-associative learning. This is distinct from other reversal learning paradigms where the same sensory modality is used and might explain why it takes mice longer to break the initial association and learn the new contingency.

On the neuronal level, we observed differences in MGB activity between initial learning and after reward contingencies were reversed. As shown in Fig. 4b and Supplementary Fig. 12-14, the global as well as local network structures during stimulus and delay periods in the naive stage after changes in stimulus contingency did not fully go back to the original state of the naive stage of initial learning, while the network structures changed in a similar manner from naive to expert stages in initial learning and after stimulus reversal. Nevertheless, it is currently not clear if and how this effect at the level of MGB might contribute to reversal learning where enhanced learning rates are observed upon reversal. However, reversal learning itself was not

the focus of our study and we have updated the wording throughout the text and conclusions about reversal learning as proposed by the reviewer (see also comment to reviewer 1). Nevertheless, since the paradigm itself relies on reversing cross-modal stimulus contingencies upon reward-associative learning, we would still prefer to talk about the 'Initial learning' and 'Reversal learning' periods in the text.

References

- Banerjee, A., Parente, G., Teutsch, J., Lewis, C., Voigt, F. F., & Helmchen, F. (2020). Value-guided remapping of sensory cortex by lateral orbitofrontal cortex. *Nature*, *585*(7824), 245–250. <https://doi.org/10.1038/s41586-020-2704-z>
- Betzel, R. F., Wood, K. C., Angeloni, C., Neimark Geffen, M., & Bassett, D. S. (2019). Stability of spontaneous, correlated activity in mouse auditory cortex. *PLoS Computational Biology*, *15*(12), e1007360. <https://doi.org/10.1371/journal.pcbi.1007360>
- Chen, S., Liu, Y., Wang, Z. A., Colonell, J., Liu, L. D., Hou, H., Tien, N.-W., Wang, T., Harris, T., Druckmann, S., Li, N., & Svoboda, K. (2024). Brain-wide neural activity underlying memory-guided movement. *Cell*, *187*(3), 676–691.e16. <https://doi.org/10.1016/j.cell.2023.12.035>
- Chuong, A. S., Miri, M. L., Busskamp, V., Matthews, G. A. C., Acker, L. C., Sørensen, A. T., Young, A., Klapoetke, N. C., Henninger, M. A., Kodandaramaiah, S. B., Ogawa, M., Ramanlal, S. B., Bandler, R. C., Allen, B. D., Forest, C. R., Chow, B. Y., Han, X., Lin, Y., Tye, K. M., ... Boyden, E. S. (2014). Noninvasive optical inhibition with a red-shifted microbial rhodopsin. *Nature Neuroscience*, *17*(8), 1123–1129. <https://doi.org/10.1038/nn.3752>
- Grant, R. I., Doncheck, E. M., Vollmer, K. M., Winston, K. T., Romanova, E. V., Siegler, P. N., Holman, H., Bowen, C. W., & Otis, J. M. (2021). Specialized coding patterns among dorsomedial prefrontal neuronal ensembles predict conditioned reward seeking. *ELife*, *10*. <https://doi.org/10.7554/eLife.65764>
- La Terra, D., Bjerre, A.-S., Rosier, M., Masuda, R., Ryan, T. J., & Palmer, L. M. (2022). The role of higher-order thalamus during learning and correct performance in goal-directed behavior. *ELife*, *11*, e77177. <https://doi.org/10.7554/eLife.77177>
- Otis, J. M., Zhu, M., Namboodiri, V. M. K., Cook, C. A., Kosyk, O., Matan, A. M., Ying, R., Hashikawa, Y., Hashikawa, K., Trujillo-Pisanty, I., Guo, J., Ung, R. L., Rodriguez-Romaguera, J., Anton, E. S., & Stuber, G. D. (2019). Paraventricular Thalamus Projection Neurons Integrate Cortical and Hypothalamic Signals for Cue-Reward Processing. *Neuron*, *103*(3), 423–431.e4. <https://doi.org/10.1016/j.neuron.2019.05.018>
- Petty, G. H., & Bruno, R. M. (2024). Attentional modulation of secondary somatosensory and visual thalamus of mice. In *bioRxiv* (p. 2024.03.22.586242). <https://doi.org/10.1101/2024.03.22.586242>
- Taylor, J. A., Hasegawa, M., Benoit, C. M., Freire, J. A., Theodore, M., Ganea, D. A., Innocenti, S. M., Lu, T., & Gründemann, J. (2021). Single cell plasticity and population coding stability in auditory thalamus upon associative learning. *Nature Communications*, *12*(1), 1–14. <https://doi.org/10.1038/s41467-021-22421-8>
- Toader, A. C., Regalado, J. M., Li, Y. R., Terceros, A., Yadav, N., Kumar, S., Satow, S., Hollunder, F., Bonito-Oliva, A., & Rajasethupathy, P. (2023). Anteromedial thalamus gates the selection and stabilization of long-term memories. *Cell*, *186*(7), 1369–1381.e17. <https://doi.org/10.1016/j.cell.2023.02.024>
- Vollmer, K. M., Green, L. M., Grant, R. I., Winston, K. T., Doncheck, E. M., Bowen, C. W., Paniccchia, J. E., Clarke, R. E., Tiller, A., Siegler, P. N., Bordieanu, B., Siemsen, B. M., Denton, A. R., Westphal, A. M., Zhou, T. C., Rinker, J. A., McGinty, J. F., Scofield, M. D., & Otis, J. M. (2022). An opioid-gated thalamoaccumbal circuit for the suppression of reward seeking in mice. *Nature Communications*, *13*(1), 1–16. <https://doi.org/10.1038/s41467-022-34517-w>

REVIEWERS' COMMENTS

Reviewer #1 (Remarks to the Author):

The paper is simply fantastic, and the authors did a great job responding to my comments. I wholeheartedly support publication.

Reviewer #1 (Remarks on code availability):

README file does not currently have directions for running code, information about data architecture, etc.

Reviewer #2 (Remarks to the Author):

The concerns I previously brought up have been addressed with reasonable responses. The authors' efforts are appreciated.

A typo was found at Supplement Methods line 50: MGB (AP: 3.2, ML: -2, DV: -3.0/-3.3 mm). should be AP:-3.2 ML:+-2 ?

Reviewer #3 (Remarks to the Author):

The manuscript is now fine for publication.

REVIEWERS' COMMENTS

Reviewer #1 (Remarks to the Author):

The paper is simply fantastic, and the authors did a great job responding to my comments. I wholeheartedly support publication.

Thank you!

Reviewer #1 (Remarks on code availability):

README file does not currently have directions for running code, information about data architecture, etc.

Thank you! We have updated the README file with directions for running code and information about the data architecture.

[https://gin.g-](https://gin.g-node.org/GrundemannLab/2024_Hasegawa_Huang_et_al_CODE/src/master/README.md)

[node.org/GrundemannLab/2024_Hasegawa_Huang_et_al_CODE/src/master/README.md](https://gin.g-node.org/GrundemannLab/2024_Hasegawa_Huang_et_al_CODE/src/master/README.md)

Reviewer #2 (Remarks to the Author):

The concerns I previously brought up have been addressed with reasonable responses. The authors' efforts are appreciated.

A typo was found at Supplement Methods line 50: MGB (AP: 3.2, ML: -2, DV: -3.0/-3.3 mm). should be AP:-3.2 ML:+-2 ?

We have updated the typo.

Reviewer #3 (Remarks to the Author):

The manuscript is now fine for publication.

Thank you!